# FtsK is critical for the assembly of the unique divisome complex of the FtsZ-less *Chlamydia trachomatis*

**McKenna Harpring[1], Junghoon Lee[2], Guangming Zhong[3], Scot P Ouellette[2], John V Cox[1]***

[1]Department of Microbiology, Immunology, and Biochemistry. University of Tennessee Health Science Center, Memphis, United States; [2]Department of Pathology, Microbiology, and Immunology, University of Nebraska Medical Center, Omaha, United States; [3]Department of Microbiology, Immunology, and Molecular Genetics, University of Texas Health San Antonio, San Antonio, United States

## eLife Assessment

Understanding how the divisome is assembled in Chlamydia trachomatis, a bacterial pathogen, is crucial since this bacterium has a non-canonical cell wall and lacks the master regulator of cell division, FtsZ. This **important** study shows that a DNA translocase, FtsK, is an early and essential component of the *Chlamydia trachomatis* divisome. The evidence presented is **convincing**, leveraging the elegant use of genetics and fluorescence microscopy. As this role of FtsK is distinct relative to most other bacteria, these findings should be of significant interest to bacterial cell biologists.

*For correspondence:
jcox@uthsc.edu

**Competing interest:** The authors declare that no competing interests exist.

**Abstract** *Chlamydia trachomatis* serovar L2 (*Ct),* an obligate intracellular bacterium that does not encode FtsZ, divides by a polarized budding process. In the absence of FtsZ, we show that FtsK, a chromosomal translocase, is critical for divisome assembly in *Ct*. Chlamydial FtsK forms discrete foci at the septum and at the base of the progenitor mother cell, and our data indicate that FtsK foci at the base of the mother cell mark the location of nascent divisome complexes that form at the site where a daughter cell will emerge in the next round of division. The divisome in *Ct* has a hybrid composition, containing elements of the divisome and elongasome from other bacteria, and FtsK is recruited to nascent divisomes prior to the other chlamydial divisome proteins assayed, including the PBP2 and PBP3 transpeptidases, and MreB and MreC. Knocking down FtsK prevents divisome assembly in *Ct* and inhibits cell division and septal peptidoglycan synthesis. We further show that MreB does not function like FtsZ and serve as a scaffold for the assembly of the *Ct* divisome. Rather, MreB is one of the last proteins recruited to the chlamydial divisome, and it is necessary for the formation of septal peptidoglycan rings. Our studies illustrate the critical role of chlamydial FtsK in coordinating divisome assembly and peptidoglycan synthesis in this obligate intracellular bacterial pathogen.

## Introduction

Most bacteria divide by a highly conserved process termed binary fission, which occurs through the symmetric division of the parental cell into two daughter cells (*Harpring and Cox, 2023*). However, *Chlamydia trachomatis* serovar L2 (*Ct*), a coccoid, gram-negative, obligate intracellular bacterium divides by a polarized cell division process characterized by an asymmetric expansion of the membrane

from one pole of a coccoid cell resulting in the formation of a nascent daughter cell (*Abdelrahman et al., 2016*; *Ouellette et al., 2022*).

*Ct* undergoes a biphasic developmental cycle during infection. Non-replicating and infectious elementary bodies (EBs) bind to and are internalized by target cells. Following internalization, EBs within a membrane vacuole, termed the inclusion, differentiate into replicating reticulate bodies (RBs). After replication, RBs undergo secondary differentiation into EBs, which are released from cells to initiate another round of infection (*Abdelrahman and Belland, 2005*).

In evolving to obligate intracellular dependence, *Ct* has eliminated several gene products essential for cell division in other bacteria, including the central coordinator of divisome formation, FtsZ (*Stephens, 1998*; *Ouellette et al., 2020*). This tubulin-like protein forms filaments that associate to form a ring at the division plane (*Barrows and Goley, 2021*), which serves as a scaffold for the assembly of the other components of the bacterial divisome that regulate the processes of septal peptidoglycan (PG) synthesis and chromosomal translocation. Of the twelve divisome proteins shown to be essential for cell division in the model gammaproteobacterial organism, *E. coli, Ct* encodes homologs of FtsK, a chromosomal translocase *Ouellette et al., 2012*; FtsQLB, regulators of septal PG synthesis (*Ouellette et al., 2015*; *Kaur and Lynn, 2022*), FtsW, a septal transglycosylase (*Putman et al., 2019*), and penicillin-binding protein 3 (PBP3/FtsI), a septal transpeptidase (*Ouellette et al., 2012*).

In addition to the divisome, rod-shaped bacteria employ another multiprotein complex, the elongasome, which directs sidewall PG synthesis (*Liu et al., 2020*) necessary for cell lengthening and the maintenance of cell shape prior to division. Although *Ct* is a coccoid organism, it encodes several elongasome proteins, including MreB, MreC, RodA, RodZ, and penicillin-binding protein 2 (PBP2), a sidewall transpeptidase (*Ouellette et al., 2012*; *Ouellette et al., 2014*; *Cox et al., 2020*). The actin-like protein MreB is essential for cell division (*Ouellette et al., 2012*; *Abdelrahman et al., 2016*) and forms septal rings in *Ct* (*Kemege et al., 2015*; *Liechti et al., 2016*; *Lee et al., 2020*). These observations led to the proposal that MreB replaces FtsZ in *Ct* and serves as a scaffold necessary for the assembly of the chlamydial divisome (*Lee et al., 2020*).

While inhibitor studies suggest that chlamydial cell division is dependent upon elements of the divisome and elongasome from other organisms (*Ouellette et al., 2012*; *Abdelrahman et al., 2016*; *Cox et al., 2020*), the composition and ordered assembly of the chlamydial divisome and its distribution during polarized budding are undefined. We hypothesized that FtsK, a chromosomal translocase, serves a critical function in regulating the division process of *Ct,* given previous observations demonstrating it interacts with elements of both the elongasome and divisome (*Ouellette et al., 2012*). We show here that FtsK is critical for the assembly of the hybrid divisome complex of *Ct* and that MreB does not serve as a scaffold necessary for the assembly of the chlamydial divisome. Rather, chlamydial MreB associates with this hybrid divisome complex late in the chlamydial divisome assembly process, and MreB filament formation is necessary for the formation of septal PG rings. Therefore, our data identify FtsK as a key regulator of the cell division process of *Ct*.

## Results
### FtsK forms foci in *Ct* that mark the location of divisome complexes

In the *E. coli* linear divisome assembly pathway (*Du and Lutkenhaus, 2017*), FtsK is the first protein downstream of FtsZ encoded by *Ct* (*Figure 1A*). In other organisms, FtsK is uniformly distributed at the septum of dividing cells (*Yu et al., 1998*; *Wang et al., 2006*; *Veiga and Pinho, 2017*). To investigate the localization of FtsK during cell division in *Ct*, HeLa cells were infected with *Ct*. To overcome the challenges associated with assessing cell morphologies in densely packed inclusions in infected cells, we analyzed FtsK localization in *Ct* derived from lysates of infected HeLa cells at 21 hr post-infection (hpi) as described previously (*Ouellette et al., 2022*). *Ct* were stained with an antibody against the chlamydial major outer membrane protein (MOMP) and an antibody that recognizes endogenous FtsK. Blotting analysis revealed that this FtsK antibody recognizes a single protein with the predicted molecular mass of FtsK (*Figure 1—figure supplement 1A*). Our results showed that, unlike FtsK in other organisms, chlamydial FtsK accumulates in discrete foci in the membrane of coccoid cells (*Figure 1B*). In cell division intermediates, FtsK localized in foci at the septum, foci at the septum and at the base of the progenitor mother cell, or foci at the base of the progenitor mother cell

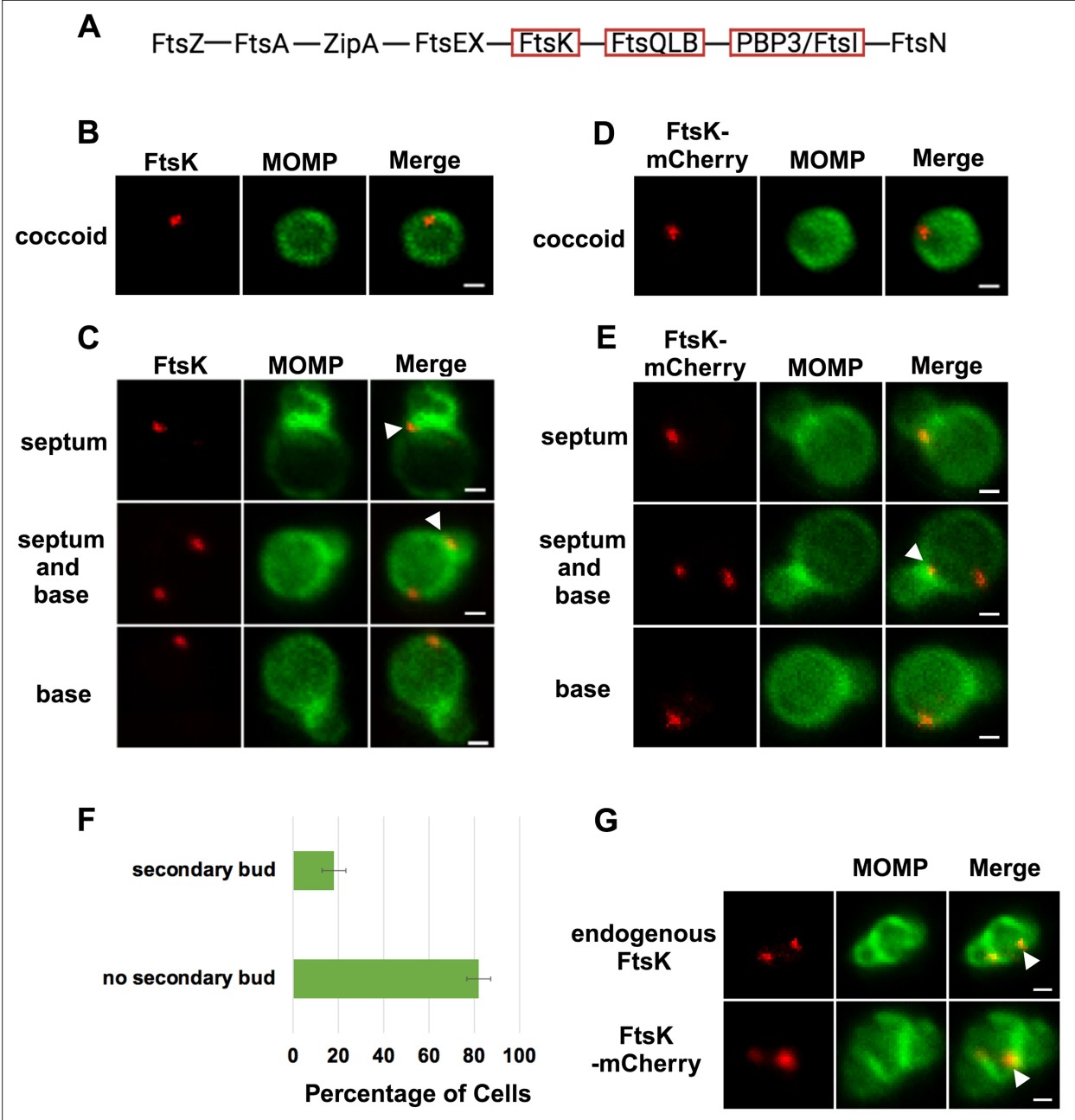

**Figure 1.** Localization of FtsK and FtsK-mCherry in coccoid cells and in dividing *Ct*. (**A**) The linear divisome assembly pathway of *E. coli* is shown. *Chlamydia trachomatis* (*Ct*) encodes the divisome proteins boxed in red. HeLa cells were infected with *Ct* L2 and reticulate bodies (RBs) were prepared at 21 hpi. The cells were fixed and stained with major outer membrane protein (MOMP) (green) and FtsK (red) antibodies. The distribution of FtsK in (**B**) coccoid cells and in (**C**) cell division intermediates that had not initiated secondary bud formation is shown. Bars are 1 μm. HeLa cells were infected with *Ct* transformed with the pBOMB4 -Tet (-GFP) plasmid encoding FtsK-mCherry. The fusion was induced with 10 nM aTc for 1 hr. and RBs were prepared from infected HeLa cells at 21 hpi and stained with MOMP antibodies (green). The distribution of MOMP relative to the mCherry fluorescence (**D**) in coccoid cells and in (**E**) cell division intermediates that had not initiated secondary bud formation is shown. Bars are 1 μm. Arrowheads in C and E denote foci of FtsK above or below the MOMP-stained septum. (**F**) HeLa cells were infected with *Ct* L2. At 21 hpi, the cells were harvested and RBs were stained with MOMP antibodies. The number of dividing cells that had initiated secondary bud formation was quantified in 150 cells. Three independent replicates were performed, and the values shown are the average of the three replicates. (**G**) Endogenous FtsK and FtsK-mCherry accumulate in foci at the septum of secondary buds (marked with arrowheads).

The online version of this article includes the following figure supplement(s) for figure 1:

**Figure supplement 1.** Blotting analysis of endogenous FtsK and FtsK-mCherry, and the effect of FtsK-mCherry overexpression on the production of infectious *Ct*.

only (*Figure 1C*). The chlamydial FtsK foci observed during cell division were not uniformly distributed at the septum, rather septal foci of FtsK were restricted to one side of the MOMP-stained septum. In addition, the FtsK foci were often above or below (marked with arrowheads in *Figure 1C*) the MOMP-stained septum. Similar analyses were performed using *Ct* transformed with the pBOMB4-Tet (-GFP) plasmid encoding FtsK with a C-terminal mCherry tag. The expression of this mCherry fusion is under the control of an anhydrotetracycline (aTc)-inducible promoter. HeLa cells were infected with the transformant, and the expression of the fusion was induced by the addition of 10 nM aTc to the media of infected cells at 19 hpi. The induced cells were harvested at 21 hpi and a lysate was prepared from *Ct* present in the infected cells. Blotting analysis of the lysate with mCherry antibodies revealed that a single protein with the predicted molecular mass of FtsK-mCherry was present in the lysate (*Figure 1—figure supplement 1B*). Imaging analyses of *Ct* in the lysate revealed that like endogenous FtsK, FtsK-mCherry accumulated in foci in coccoid cells (*Figure 1D*), and in division intermediates, it localized in foci at the septum, foci at the septum and at the base of the progenitor mother cell, or foci at the base of the progenitor mother cell only (*Figure 1E*). The foci of FtsK-mCherry, like endogenous FtsK, were often offset relative to the plane defined by MOMP staining at the septum (arrowhead in *Figure 1E*). Inclusion forming unit (IFU) assays demonstrated that overexpression of the FtsK-mCherry fusion had no effect on chlamydial developmental cycle progression and the production of infectious EBs (*Figure 1—figure supplement 1C*). While it is possible that the population of FtsK at the base of the mother cell is a remnant of FtsK from a previous division, ~20% of dividing cells have a secondary bud (*Figure 1F*), and FtsK and FtsK-mCherry accumulate in foci at the base of secondary buds (arrowheads in *Figure 1G*), suggesting that the population of FtsK at the base of the mother cell corresponds to a nascent divisome complex that forms at the site where the daughter cell will arise in the next round of division.

To investigate the distribution of other putative chlamydial divisome components during budding, we transformed *Ct* with plasmids encoding PBP2, PBP3, or MreC with an N-terminal mCherry tag. The PBP2, PBP3, and MreC fusions were induced by the addition of 10 nM aTc to infected cells at 19 hpi, and the induced cells were harvested at 21 hpi and stained with MOMP antibodies. Imaging analyses revealed that the PBP2, PBP3, and MreC fusions accumulated in foci in coccoid cells (*Figure 2A*), and in cell division intermediates, the fusions accumulated in foci at the septum, in foci at the septum and at the base of the progenitor mother cell, or in foci only at the base of the progenitor mother cell (*Figure 2B*). Similar analyses with an MreB_6xHis fusion (*Lee et al., 2020*) revealed that MreB exhibited a similar localization profile (*Figure 2A and B*). Each of these fusions, like FtsK, were restricted to one side and were often slightly above or slightly below (marked with arrowheads in *Figure 2B*) the MOMP-stained septum in dividing cells. Blotting analyses revealed that mCherry antibodies primarily detected single species with the predicted molecular mass of the PBP2, PBP3, and MreC fusions in lysates prepared from induced cells (*Figure 2—figure supplement 1A*), and IFU assays demonstrated that the aTc induced overexpression of the PBP2, PBP3, and MreC fusions had no effect on the developmental cycle progression of *Ct* (*Figure 2—figure supplement 1B*). Like FtsK, the foci of the PBP2, PBP3, MreC and MreB fusions were also detected at the base of secondary buds (*Figure 2—figure supplement 1C*). Quantification of the localization profiles of endogenous FtsK and the various fusion proteins revealed that the distribution profile of FtsK-mCherry accurately reflected the distribution of endogenous FtsK (*Figure 2C*). Furthermore, a greater percentage of FtsK was associated with the base of dividing cells (including cells with septum and base, and cells with base alone) suggesting that FtsK associates with nascent divisomes at the base of dividing cells prior to the other putative divisome proteins assayed. Finally, this analysis suggested that MreB associated with nascent divisomes at the base of dividing cells after mCherry-PBP2 and mCherry-PBP3 (marked with # in *Figure 2C*). The localization profiles of the chlamydial divisome proteins (*Figures 1 and 2*) likely reflect the assembly of divisome complexes at the septum and at the base of the progenitor mother cell, and the disassembly of the septal divisome when divisome proteins are only present at the base of the mother cell.

Since it was possible that the localization profiles of the mCherry-PBP2 and mCherry-PBP3 fusions were at least in part due to their induced over-expression, we performed similar studies with rabbit antibodies generated against peptides derived from PBP2 or PBP3 (*Ouellette et al., 2012*). Blotting analyses (*Figure 2—figure supplement 2A*) with these antibodies revealed that they recognized mCherry-PBP2 and mCherry-PBP3 in *Ct* lysates, and immunofluorescent staining with the PBP2 and PBP3-specific antibodies (*Figure 2—figure supplement 2B*) completely overlapped the

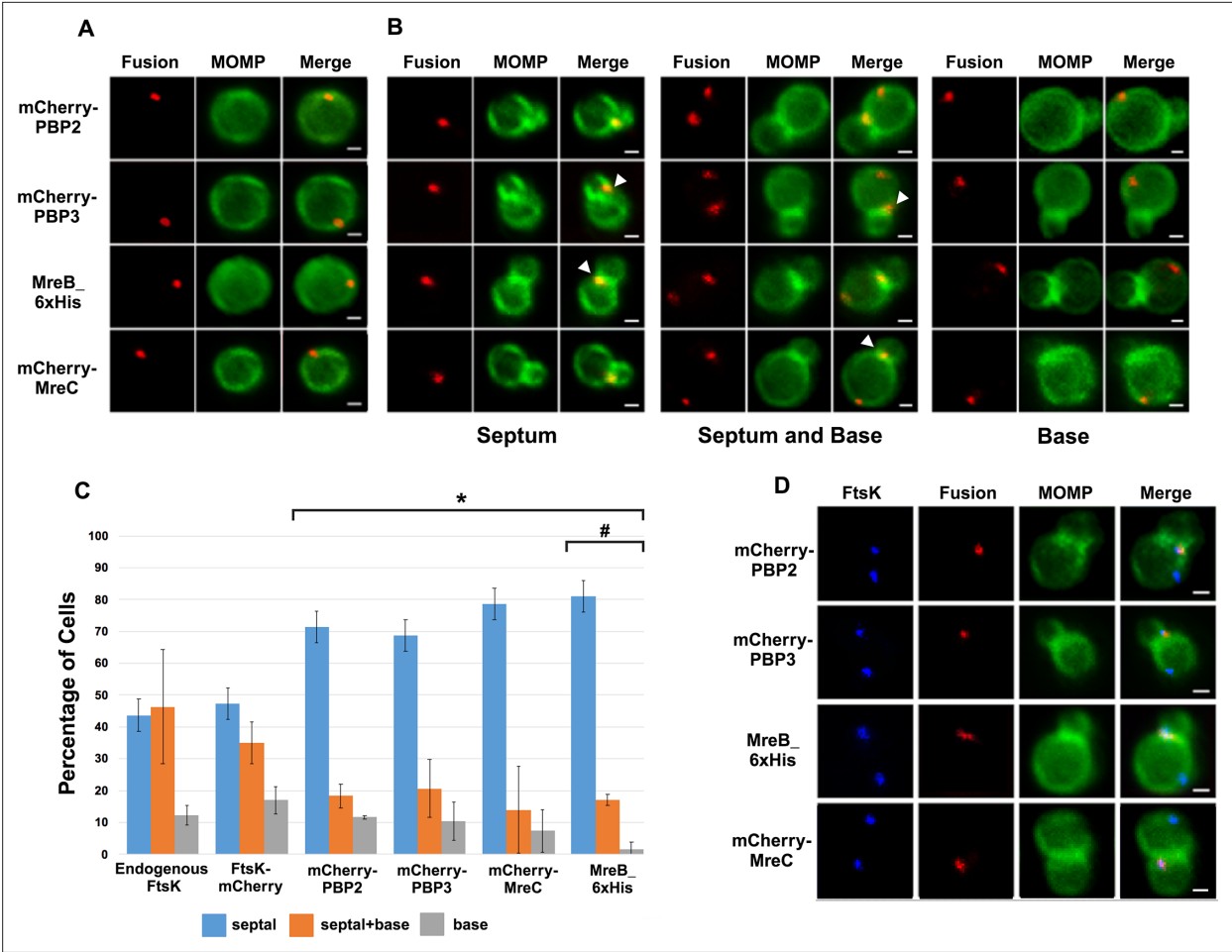

**Figure 2.** Localization of the fusions of PBP2, PBP3, MreC, and MreB in *Ct* and a comparison of their localization to the localization of endogenous FtsK. HeLa cells were infected with *Ct* transformed with PBP2, PBP3, or MreC with an N-terminal mCherry tag, or with *Ct* transformed with a MreB_6xHis fusion (Lee, 2020). Each of the fusions was induced by adding 10 nM aTc to the media at 17 hpi. Lysates were prepared at 21 hpi and the cells were fixed and stained with a major outer membrane protein (MOMP) antibody. The distribution of the mCherry fluorescence in (**A**) coccoid cells and in (**B**) dividing cells that had not initiated secondary bud formation is shown. Cells expressing the MreB_6xHis fusion were stained with rabbit anti-6x His antibody (red) and MOMP antibodies (green). Dividing cells with foci at the septum, foci at the septum, and foci at the base of the mother cell, or foci at the base alone are shown for each of the fusions. Arrowheads in B denote foci of the divisome proteins above or below the plane of the MOMP-stained septum. (**C**) HeLa cells were infected with *Ct* L2. Alternatively, cells were infected with *Ct* that were transformed with FtsK-mCherry, mCherry-PBP2, mCherry-PBP3, mCherry-MreC, or MreB_6xHis. The cells expressing the fusions were induced with aTc for 1.5 hr then fixed at 21 hpi and the distribution of endogenous FtsK, or the mCherry fluorescence in cells inducibly expressing the mCherry fusions, or the distribution of MreB in cells where the MreB_6xHis fusion was inducibly expressed was quantified. A small fraction (2%) of dividing cells contained multiple foci of each of the divisome proteins at the septum and/or the base of the cell. Since these cells were relatively rare, we included these cells with cells that contained a single foci in the quantification. The localization profiles were quantified in 100 cells. Three independent replicates were performed, and the values shown are the average of the three replicates. Chi-squared analysis revealed that the localization profiles of endogenous FtsK and FtsK-mCherry are not statistically different from each other, but they are statistically different than the PBP2, PBP3, MreC, and MreB localization profiles (*p<0.009). The localization profile of the MreB fusion is also statistically different than the localization profiles of the mCherry fusions of PBP2 and PBP3 (#p=0.05). (**D**) Hela cells were infected with *Ct* transformed with PBP2, PBP3, or MreC with a N-terminal mCherry tag, or with *Ct* transformed with an MreB_6xHis fusion (***Lee et al., 2020***). The fusions were induced by adding 10 nM aTc to the media at 17 hpi. The cells were harvested at 21 hpi and *Ct* were harvested and stained with FtsK and MOMP antibodies. The cells expressing the MreB fusion were stained with FtsK, MOMP, and 6xHis antibodies. Imaging analyses revealed that FtsK was present in foci at the septum and in foci at the base in these cells, while each of the fusions was only detected at the septum where they overlapped the distribution of septal FtsK (Bars are1 µm).

The online version of this article includes the following figure supplement(s) for figure 2:

**Figure supplement 1.** Characterization of the fusions of PBP2, PBP3, MreC and MreB expressed in chlamydial cells.

**Figure supplement 2.** Characterization mCherry-PBP2 and mCherry-PBP3 expressed in *Ct*.

**Figure supplement 3.** Characterization of endogenous PBP2 and PBP3 localization in coccoid cells and in dividing *Ct*.

*Figure 2 continued on next page*

*Figure 2 continued*

**Figure supplement 4.** mCherry tagged versions of PBP2, PBP3, and MreC, and the 6xHis tagged version of MreB accumulate in foci at the septum and base or the base alone in some dividing *Ct* that overlap the localization of endogenous FtsK.

**Figure supplement 5.** MreC rings in (**A**) dividing Ct and in (**B**) coccoid Ct. Bars are 1 μm.

mCherry fluorescence in cells when the mCherry PBP2 and PBP3 fusions were inducibly expressed in *Ct*. Imaging analyses with the antibodies that recognize endogenous PBP2 and endogenous PBP3 indicated that these antisera detected foci in coccoid cells, and in cell division intermediates, the PBP2 and PBP3 antibodies detected foci at the septum, foci at the septum and at the base of the mother cell, or foci at the base alone (*Figure 2—figure supplement 3A*). Quantification revealed that the localization profiles of endogenous PBP2 and PBP3 in division intermediates (*Figure 2—figure supplement 3B*) were not statistically different than the localization profiles observed for the mCherry fusions of PBP2 and PBP3 (*Figure 2C*).

The quantification in *Figure 2C* suggested that FtsK is recruited to nascent divisomes that form at the base of dividing cells prior to the other divisome components assayed. This hypothesis was tested by staining cells expressing the PBP2, PBP3, MreC, or MreB fusions with antibodies that recognize endogenous FtsK. Imaging analyses revealed that in a subset of cells, FtsK was detected in foci at the septum and at the base of dividing cells, while each of the fusions was only detected at the septum where they overlapped the distribution of septal FtsK (*Figure 2D*), indicating that FtsK is recruited to nascent divisomes at the base of the cell prior to the other divisome components assayed. Additional analyses revealed that FtsK also overlapped the distribution of the PBP2, PBP3, MreC, and MreB fusions following their appearance at the base of dividing cells (*Figure 2—figure supplement 4A*). Quantification revealed the percentage of cells in which FtsK overlapped the distribution of each of the fusions at the septum, at the septum and the base, and at the base alone in division intermediates (*Figure 2—figure supplement 4B*).

## MreB filament formation is not required for foci formation by FtsK, PBP2, and PBP3

MreB was one of the last components that associated with nascent divisomes forming at the base of the progenitor mother cell (*Figure 2C*). To investigate whether MreB filament formation was required for the formation of foci by the other chlamydial divisome components, HeLa cells were infected with *Ct* transformed with the FtsK, PBP2, PBP3, MreC, or MreB fusions, and the fusions were induced by adding 10 nM aTc to the media of the infected cells at 20 hpi for 1 hr. During the induction period, cells were incubated in the absence (*Figure 3B and D*) or presence (*Figure 3C and E*) of the MreB inhibitor, A22, which inhibits MreB filament formation (*Bean et al., 2009*). RBs were harvested at 21 hpi and stained with the appropriate antibodies to assess the effect of A22 on the cellular distribution of the fusions. As previously shown (*Ouellette et al., 2012*; *Cox et al., 2020*), A22 inhibits chlamydial budding and most cells in the population were coccoid following A22 treatment (*Figure 3A*). Furthermore, approximately 50% of the untreated control cells were coccoid, which is consistent with prior estimates of the number of non-dividing RBs at this stage of the developmental cycle (*Lee et al., 2018*), indicating that our lysis procedure does not lead to a bias in the number of non-dividing coccoid cells in the population. MreB in coccoid cells adopted a diffuse pattern of localization following A22 treatment (*Figure 3*). A22 also had a statistically significant effect on the percent of coccoid cells containing MreC foci, but it did not affect the ability of FtsK, PBP2, or PBP3 to form foci in coccoid cells (*Figure 3D and E*). These data indicate that MreB filaments do not function as a scaffold that is necessary for the assembly of all divisome components in *Ct*.

## Effect of *ftsk* and *pbp2* knockdown on cell division and divisome assembly in *Ct*

To further investigate the mechanisms that regulate divisome assembly in *Ct*, we inducibly repressed the expression of *ftsK* or *pbp2* using CRISPRi technology, which has been used to inducibly repress the expression of genes in *Ct* (*Ouellette et al., 2021*). CRISPRi employs a constitutively expressed crRNA that targets an inducible dCas enzyme (dCas12) to specific genes where it binds but fails to

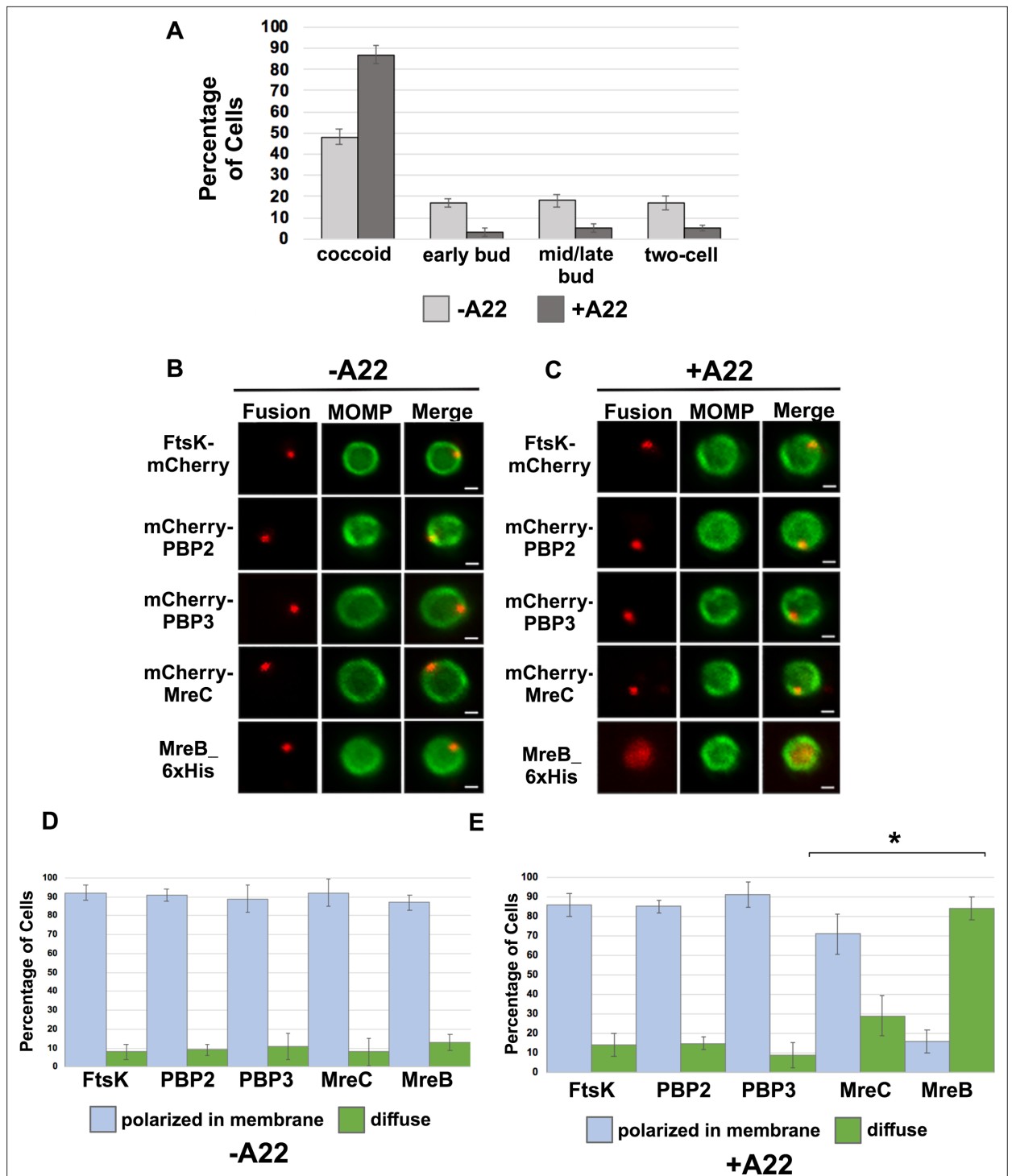

**Figure 3.** Effect of A22 on cell morphology and foci formation by mCherry tagged FtsK, PBP2, PBP3 and MreC, and foci formation by MreB_6xHis. (**A**) HeLa cells infected with *Chlamydia trachomatis* (*Ct*) were treated with 75 µM A22 for 1 hr. Control cells were not treated with A22. Lysates were prepared form A22 treated and untreated cells at 21 hpi and the number of coccoid and dividing cells in the population were quantified in 100 cells. Three independent replicates were performed, and the values shown are the average of the three replicates. (**B–E**) Alternatively, HeLa cells were infected with *Ct* transformed with plasmids encoding FtsK-mCherry, mCherry-PBP2, mCherry-PBP3, mCherry-MreC, or MreB-6xHis. The fusions were induced at 20 hpi with 10 nM aTc for 1 hr in the absence (**B and D**) or presence (**C and E**) of 75 µM A22. Coccoid cells prepared from the infected cells at 21 hpi were stained with major outer membrane protein (MOMP) antibodies (green). The MreB-6xHis fusion was also stained with 6xHis antibodies (red). Panel B shows the distribution of the fusions in untreated coccoid cells. Panel C illustrates the effect of A22 on the localization of the fusions in coccoid cells. Bars in B and C are 1 µm. The distribution of FtsK-mCherry, mCherry-PBP2, mCherry-PBP3, mCherry-MreC, and MreB-6xHis was quantified

*Figure 3 continued on next page*

Figure 3 continued

in (**D**) control coccoid cells and in (**E**) A22-treated cells coccoid cells (n=50) is shown. Three replicates were performed, and the values shown in D and E are the averages of the three replicates. Student T-test indicated that A22 had a statistically significant effect on the localization of MreB and MreC (*p<0.01).

cut, thus inhibiting transcription. We transformed *Ct* with the pBOMBL12CRia plasmid that constitutively expresses an *ftsK* or *pbp2*-specific crRNA, which target sequences in the *ftsK* or *pbp2* promoter regions. To determine whether *ftsK* and *pbp2* transcript levels were altered using this CRISPRi approach, dCas12 expression was induced by the addition of 5 nM aTc to the media of infected cells at 8 hpi. Control cells were not induced. Nucleic acids were isolated from induced cells and uninduced control cells at various times, and RT-qPCR was used to measure *ftsK* or *pbp2* transcript levels. This analysis revealed that the induction of dCas12 resulted in ~10 fold reduction in *ftsK* transcript levels by 15 hpi in cells expressing the *ftsK*-targeting crRNA (*Figure 4A*), and ~ eightfold reduction in *pbp2* transcript levels in cells expressing the *pbp2*-targeting crRNA (*Figure 4B*), while these crRNAs had minimal or no effect on chlamydial *euo* and *omcB* transcript levels, suggesting that the *ftsK* and *pbp2* crRNAs specifically inhibit the transcription of *ftsK* and *pbp2* (*Figure 4A and B*). To investigate the effect of *ftsK* or *pbp2* down-regulation on developmental cycle progression, dCas12 was induced by the addition of aTc to the media of infected cells at 4 hpi. Control cells were not induced. The cells were then fixed at 24 hpi and stained with MOMP and Cas12 antibodies. Imaging analysis revealed that *Ct* morphology was normal and dCas12 was undetectable in the inclusions of uninduced control cells, while foci of dCas12 were observed in the induced cells, and *Ct* in the inclusion exhibited an enlarged aberrant morphology (*Figure 4C and D*), suggesting that the inducible knockdown of *ftsK* or *pbp2* blocks chlamydial cell division. In additional studies, we induced dCas12 at 17 hpi in cells expressing the *ftsK* or *pbp2*-targeting crRNAs. Lysates were prepared and the cells were fixed at 21 hpi, and localization studies revealed that foci of endogenous FtsK and PBP2 were almost undetectable when *ftsK* or *pbp2* were transiently knocked down using this CRISPRi approach (*Figure 4E*).

To assess whether the knockdown of *ftsK* or *pbp2* arrests *Ct* division at a specific stage of polarized budding, HeLa cells were infected with *Ct* transformed with the pBOMBL12CRia plasmids encoding the *ftsK* or *pbp2*-targeting crRNAs. At 17 hpi, dCas12 was induced by the addition of 20 nM aTc to the media. Control cells were not induced. RBs were harvested from induced and uninduced control cells at 22 hpi and stained with MOMP antibodies, and imaging analyses quantified the division intermediates present in the population. These analyses revealed that >60% of the *Ct* in the uninduced controls were at various stages of polarized budding (*Figure 5A and B*), while ~90% of the cells were coccoid when *ftsK* was knocked down (*Figure 5A*), and ~85% of the cells were coccoid following *pbp2* knockdown (*Figure 5B*), suggesting that the initiation of polarized budding of *Ct* is inhibited when *ftsK* or *pbp2* are knocked down.

We next examined whether the knockdown of *ftsK* affected foci formation by PBP2 or PBP3 in coccoid cells. HeLa cells were infected with *Ct* transformed with the pBOMBL12CRia plasmid encoding the *ftsK*-targeting crRNA. At 17 hpi, dCas12 was induced by the addition of 10 nM aTc to the media. Control cells were not induced. RBs were harvested from induced and uninduced control cells at 22 hpi, and the localization of endogenous PBP2 and PBP3 in coccoid cells was assessed (*Figure 5C and D*). This analysis revealed that the number of coccoid cells with polarized foci of PBP2 and PBP3 were reduced by approximately 80% following *ftsK* knockdown (*Figure 5E*). In similar analyses, we assessed the effect of *pbp2* knockdown on the ability of FtsK and PBP3 to form foci in coccoid cells. While FtsK retained its ability to form foci in coccoid cells following *pbp2* knockdown, foci of PBP3 were almost entirely absent in *pbp2* knockdown cells (*Figure 5C and D*). Quantification of these assays revealed that FtsK is necessary for foci formation by both the PBP2 and PBP3 transpeptidases, while PBP2 is necessary for foci formation by PBP3 (*Figure 5E and F*). Our data place FtsK upstream of, and necessary for, the addition of PBP2 and PBP3 to the *Ct* divisome, and PBP2 upstream of and necessary for the addition of PBP3 to the *Ct* divisome. These results are consistent with inhibitor studies that indicated PBP2 acts upstream of PBP3 in the polarized budding process of *Ct* (*Cox et al., 2020*).

To investigate whether the catalytic activity of PBP2 is necessary to maintain its association with the *Ct* divisome, HeLa cells were infected with *Ct*, and mecillinam, an inhibitor of the transpeptidase activity of PBP2 (*Kocaoglu et al., 2015*; *Cox et al., 2020*), was added to the media of infected cells at 20 hpi. Cells incubated in the absence of mecillinam were included as a control. Mecillinam-treated

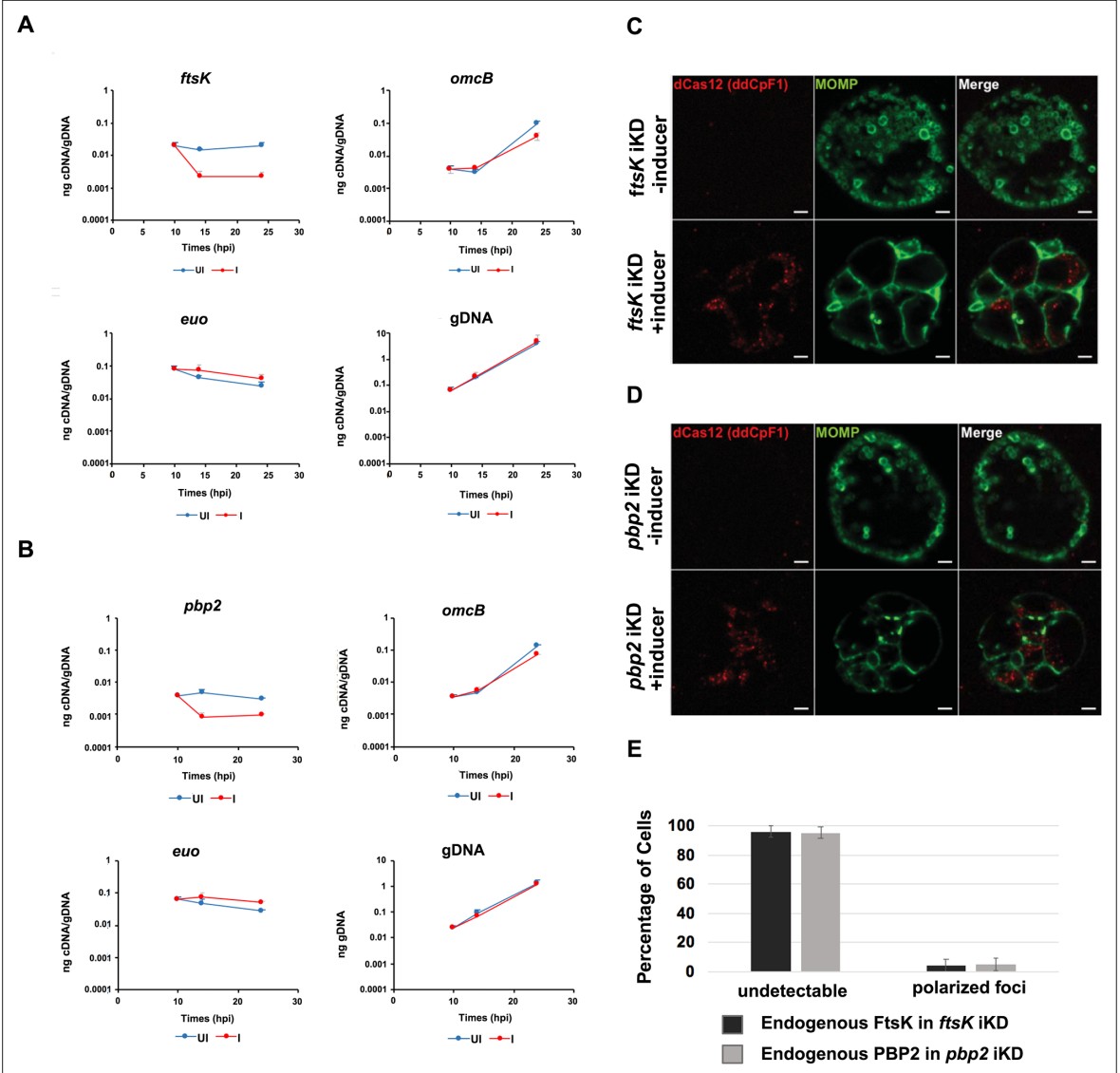

**Figure 4.** Effect of CRISPRi knock down of *ftsK or pbp2 on* gene expression, cell morphology and foci formation by FtsK and PBP2. HeLa cells were infected with *Chlamydia trachomatis* (*Ct*) transformed with the pBOMBL12CRia plasmid that constitutively expresses *ftsK* or *pbp2*-targeting crRNAs. dCas12 expression was induced by the addition of 5 nM aTc to the media of infected cells at 8 hpi. Control cells were not induced. Nucleic acids were isolated from induced cells and from uninduced controls at various times post-infection, and RT-qPCR was used to measure *ftsK* or *pbp2* transcript levels. (**A**) The induction of dCas12 resulted in ~10 fold reduction in *ftsK* transcript levels in cells expressing the *ftsK*-targeting crRNA, (**B**) and ~ eightfold reduction in *pbp2* transcript levels in cells expressing the *pbp2*-targeting crRNA, while these crRNAs had minimal or no effect on chlamydial *euo* and *omcB* transcript levels. HeLa cells were infected with *Ct* transformed pBOMBL12CRia plasmid that constitutively expresses a (**C**) *ftsK* or (**D**) *pbp2*-targeting crRNA. dCas12 expression was induced by the addition of 5 nM aTc to the media of infected cells at 8 hpi. Control cells were not induced. The infected cells were fixed at 24 hpi and stained with MOMP and Cas12 antibodies. *Ct* morphology was normal and dCas12 was undetectable in the inclusions of uninduced control cells. Foci of dCas12 were observed in induced cells, and *Ct* in the inclusion exhibited an enlarged aberrant morphology. Bars in C and D are 2 μm. (**E**) HeLa cells were infected with *Ct* transformed with the pBOMBL12CRia plasmid that constitutively expresses a *ftsK* or *pbp2*-targeting crRNA. dCas12 was induced at 17 hpi by the addition of 10 nM aTc to the media. Control cells were not induced. The cells were harvested at 21 hpi, and *Ct* were prepared and stained with antibodies that recognize that endogenous FtsK or PBP2. Quantification shows that polarized foci of FtsK and PBP2 were almost undetectable when *ftsK* or *pbp2* were transiently knocked down.

and control cells were harvested at 22 hpi and the effect of inhibiting the catalytic activity of PBP2 on the localization of FtsK, PBP2, and PBP3 was determined. As shown previously, mecillinam blocks chlamydial division (*Ouellette et al., 2012*; *Cox et al., 2020*), and most cells in the population assumed a coccoid morphology (*Figure 6A*). We then determined the localization of endogenous FtsK, PBP2, and PBP3 in drug-treated and control coccoid cells. Mecillinam treatment resulted in a ~50% reduction

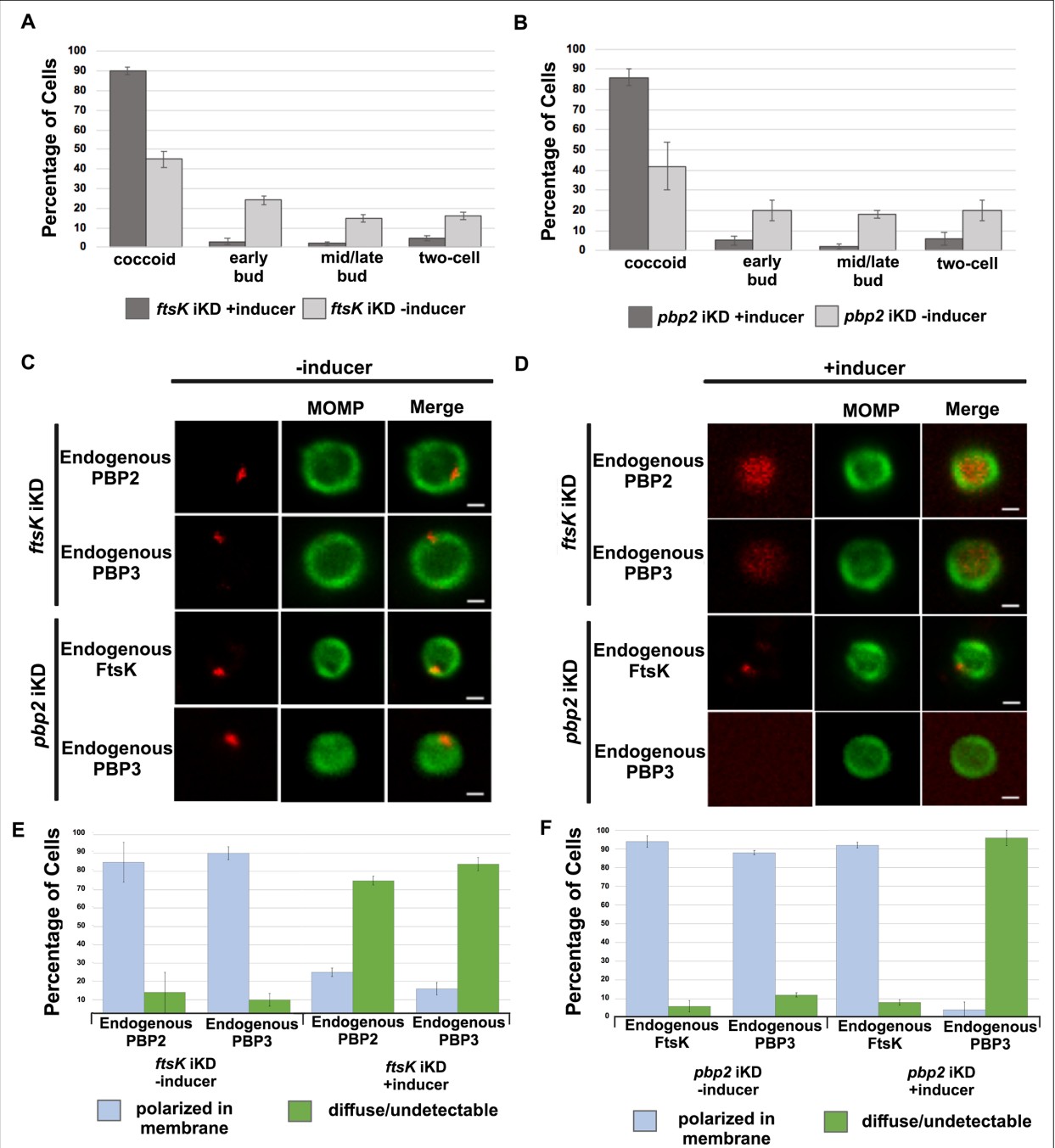

**Figure 5.** Effect of *ftsK or pbp2* inducible knockdown on cell morphology and foci formation by FtsK, PBP2, and PBP3.
HeLa cells were infected with *Ct* transformed with the pBOMBLcRia plasmid, which constitutively expresses a *ftsK* or *pbp2* crRNA and dCas12 under the control of an aTc-inducible promoter. dCas12 was induced at 17 hpi by adding 5 nM aTc to the media. In a control infection, the expression of dCas12 was not induced. Cells were harvested at 24 hpi and the morphology of *Ct* in induced and uninduced control cells was assessed in 250 cells. Three replicates were performed, and the values shown are the averages of the three replicates (**A and B**). The localization of endogenous FtsK, endogenous PBP2, and endogenous PBP3 was assessed in cells transformed with the pBOMBL-12CRia plasmid that targets *ftsK* or *pbp2*. The localization is shown in coccoid cells where dCas12 expression was (**C**) uninduced or (**D**) induced. White bars are 1 μm. (**E and F**) The localization profiles of FtsK, PBP2, and PBP3 were quantified in uninduced and induced cells. Three replicates were performed, and the values shown are the averages of the three replicates.

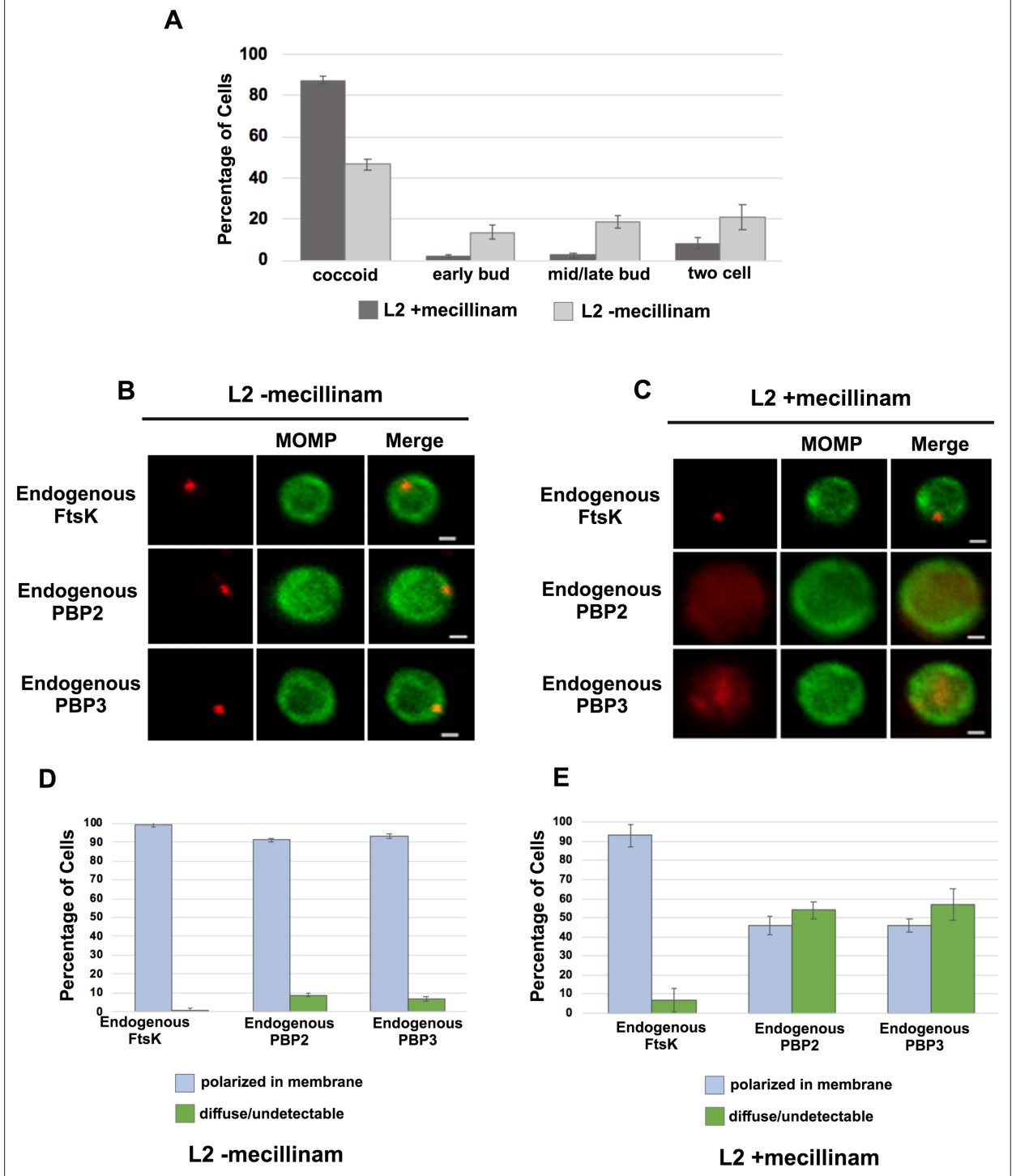

**Figure 6.** Effect of mecillinam on cell morphology and on the localization of endogenous FtsK, PBP2, and PBP3. (**A**) HeLa cells were infected with *Ct* and 20 µM mecillinam was added to the media at 17 hpi. Untreated coccoid cells were included as a control. The cells were harvested at 21 hpi and the morphology of MOMP-stained cell was assessed in 200 cells. Three replicates were performed, and the values shown are the averages of the three replicates. (**B and C**) The localization of endogenous FtsK, endogenous PBP2, and endogenous PBP3 in untreated coccoid or in mecillinam-treated coccoid cells is shown. Bars are1 µM. (**D and E**) Localization of FtsK, PBP2, and PBP3 in untreated and mecillinam-treated coccoid cells was quantified in 50 cells. Three replicates were performed, and the values shown are the averages of the three replicates.

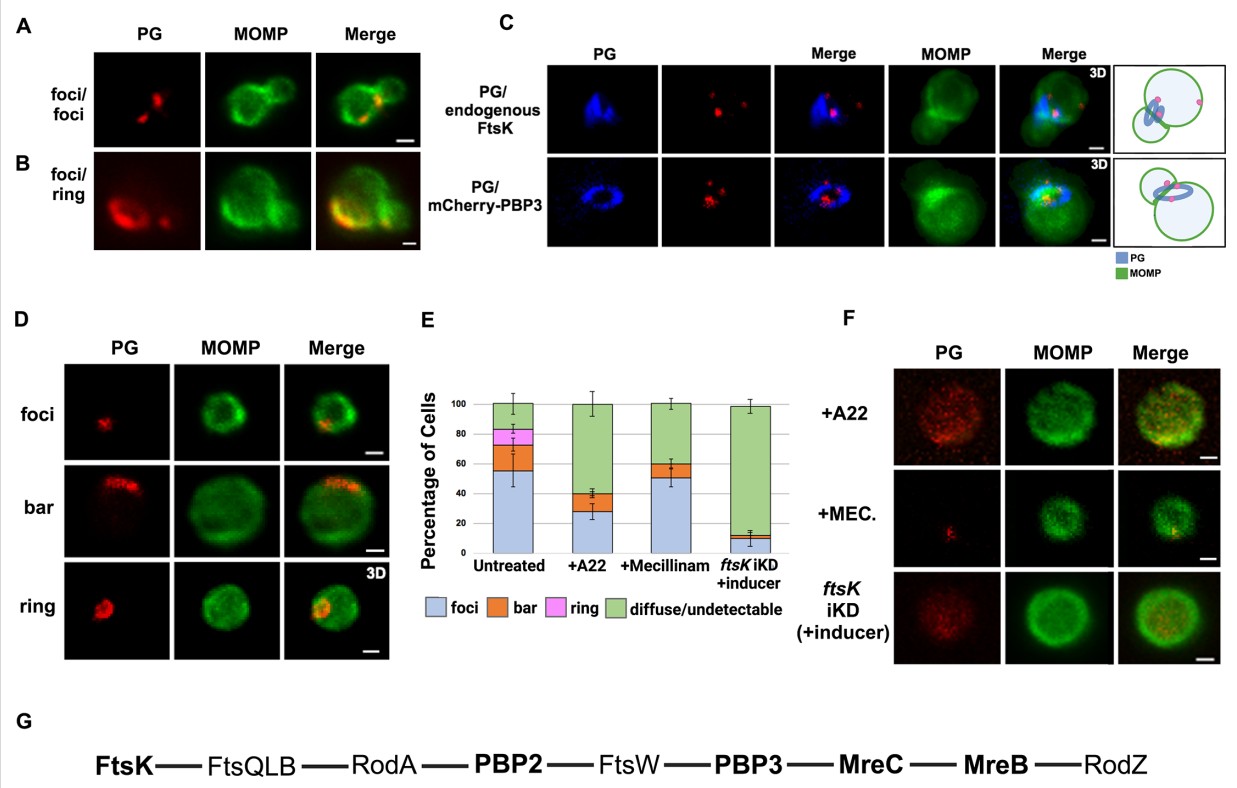

**Figure 7.** Peptidoglycan (PG) distribution in *Chlamydia trachomatis* (*Ct*). HeLa cells were infected with *Ct* L2. At 17 hpi, 4 mM ethylene-DA-DA (EDA-DA) was added to the media, the cells were harvested at 21 hpi, and the ethylene-DA-DA (EDA-DA) was click labeled and compared to the distribution of major outer membrane protein (MOMP). (**A**) Imaging analyses revealed that PG formed foci at the septum and at the base in some dividing cells. (**B**) Imaging analyses revealed that PG organization at the septum and at the base of some dividing cells differed. In this example, a PG foci was detected at the septum and a PG ring was detected at the base of a dividing cell. (**C**) The localization of click-labeled PG was compared to the localization of endogenous FtsK and mCherry-PBP3 at the septum of dividing cells. 3D projections revealed that multiple foci of each protein are associated with PG intermediates. Cartoons are included to assist the reader in visualizing the angled orientation of PG relative to the MOMP-stained septum in the dividing cells. (**D**) PG organization in untreated coccoid cells. (**E**) Quantification of PG organization in untreated coccoid cells, A22-treated coccoid cells, mecillinam-treated coccoid cells, and in coccoid cells resulting from the inducible knockdown of *ftsk*. Fifty cells were counted for each condition. Three replicates were performed and the average from the three replicates is shown. (**F**) PG organization in A22-treated and mecillinam-treated coccoid cells, and in coccoid cells resulting from the inducible knockdown of *ftsk* is shown. Bars are 1 μm. (**G**) Putative *Ct* divisome assembly pathway is shown. Proteins characterized in this study are bolded. The ordering of the remaining proteins is based on the assembly of the divisome and elongasome in *E. coli* (***Du and Lutkenhaus, 2017***; ***Liu et al., 2020***).

in the number of cells with polarized foci of PBP2 (***Figure 6B-E***). There was a similar reduction in polarized foci of PBP3 following mecillinam treatment (***Figure 6B-E***). These data indicate that the catalytic activity of PBP2 is necessary for PBP2 to efficiently associate with or maintain its association with polarized divisome complexes. Furthermore, consistent with *pbp2* knockdown studies, PBP3 association with the divisome complex is dependent on the prior addition of PBP2 to the complex, but foci formation by FtsK is unaffected when PBP2 foci are reduced in number (***Figure 6***).

## Effect of inhibitors and *ftsk* knockdown on PG organization in *Ct*

To assess the morphology of PG at the septum and the base of dividing cells, we used an EDA-DA labeling strategy (***Liechti et al., 2014***; ***Cox et al., 2020***). This approach enabled the detection of PG foci, bars, and rings in dividing *Ct* (***Liechti, 2021***). *Our* imaging analysis revealed that in some instances PG foci were detected at both the septum and at the base of the progenitor mother cell (***Figure 7A***), and in other instances, PG organization at the septum and the base differed. In the example shown (***Figure 7B***), a PG foci was detected at the septum while a PG ring was detected at the base of the mother cell. In additional analyses, we compared the localization of mCherry-PBP3 to the localization of PG in cells where the expression of this mCherry fusion had been induced by the addition of aTc to

the media. This analysis, which was restricted to PG formation at the septum of dividing cells, revealed that multiple foci of PBP3 were associated with a septal PG ring (*Figure 7C*). Furthermore, the PG ring was at an angle relative to the MOMP-stained septum. Similar analyses revealed that two foci of endogenous FtsK were associated with PG that was again at an angle relative to the MOMP-stained septum. This was true even though the PG had not fully reorganized into a ring structure (*Figure 7C*).

We then determined the effect of A22 and mecillinam on PG synthesis/organization. Since both of these drugs induce *Ct* to assume a coccoid morphology, we initially characterized PG organization in untreated coccoid cells. We detected foci, bars, or rings in ~80% of untreated coccoid cells (*Figure 7D and E*), which make up ~50% of the cells in the inclusion at this stage of the developmental cycle (*Lee et al., 2018*). Furthermore, each of these PG intermediates exhibited a polarized distribution in untreated coccoid cells (*Figure 7D*). Although we cannot rule out that continued PG synthesis and reorganization occurs in polarized division intermediates, PG rings can arise prior to any of the morphological changes that occur during the polarized division of *Ct*.

Prior studies have shown that inhibitors of MreB filament formation prevent the appearance of PG-containing structures in *Ct* (*Liechti et al., 2014*; *Ouellette et al., 2022*). To assess whether MreB filament formation is required for PG synthesis and/or PG reorganization, we infected HeLa cells with *Ct*, and EDA-DA and A22 were added to the media of infected cells at 18 hpi. The cells were harvested at 22 hpi, lysates were prepared, and PG localization was determined. These analyses revealed that PG was diffuse/undetectable in the majority of A22-treated cells, and, in those cells where PG was still detected, it could not convert into ring structures (*Figure 7E and F*). In similar experiments, we assessed the effect of mecillinam on the appearance of PG intermediates in coccoid cells. These analyses revealed that PG formed discrete foci or bars in 60% of mecillinam-treated cells (*Figure 7E and F*). However, these PG intermediates could not convert into PG rings when the transpeptidase activity of PBP2 was inhibited. Finally, we assessed PG organization in cells where *ftsK* was knocked down by inducing dCas12 in the *ftsK* knockdown strain by the addition of aTc to the media of infected cells at 17 hpi. Cells were fixed at 21 hpi, and localization studies revealed that *ftsK* knockdown had the most dramatic effect on PG localization, which was diffuse/undetectable in ~90% of the cells assayed. Inhibiting divisome assembly by knocking down *ftsK* almost entirely prevented the accumulation of all PG-containing intermediates in *Ct* (*Figure 7E and F*).

## Discussion

The results presented here provide insight into the molecular mechanisms governing the FtsZ-less polarized cell division process of *Ct*. This study is the first to document the ordered assembly of divisome proteins in *Ct* and to investigate the roles of divisome proteins in regulating PG synthesis/organization in this obligate intracellular bacterial pathogen. Our studies showed that the divisome in *Ct* is hybrid in composition, containing elements of the divisome and elongasome from other bacteria, and a putative pathway for the assembly of the novel hybrid divisome of *Ct* is shown in *Figure 7G*. Although we focused our study on a subset of the divisome and elongasome proteins that *Chlamydia* expresses (bolded in *Figure 7G*), our results support our conclusion that chlamydial budding is dependent upon a hybrid divisome complex and that FtsK is required for the assembly of this hybrid divisome. At this time, we cannot rule out that other proteins act upstream of FtsK to initiate divisome assembly in this obligate intracellular bacterial pathogen.

The domain organization of FtsK is highly conserved among bacteria. The C-terminus of the protein mediates the ATP-dependent directional translocation of the protein along DNA through interaction with KOPS (Fts**K o**rienting **p**olar **s**equences) (*Barre et al., 2001*; *Massey et al., 2006*) and stimulates XerCD-dependent recombination at the *dif* site near the chromosomal terminus to decatenate chromosomal dimers that arise from homologous recombination during DNA replication to monomers (*Blakely et al., 1993*; *Massey et al., 2006*). The C-terminus of chlamydial FtsK is 41% identical to the C-terminus of *E. coli* FtsK. The N-terminus of *E. coli* FtsK encodes four predicted transmembrane spanning helices, which are sufficient to direct the protein to the septum in dividing cells (*Yu et al., 1998*). The N-terminus of chlamydial FtsK, which is also predicted to encode four transmembrane-spanning helices, is 34% identical to *E. coli* FtsK.

Our analyses revealed a novel spatiotemporal localization pattern of FtsK during the chlamydial division process. Chlamydial FtsK forms discrete foci at the septum, foci at the septum, and at the base of the mother cell, or in foci only at the base of the mother cell (*Figure 1*). Our data indicate

that the foci at the base of the mother cell correspond to nascent divisome complexes that form prior to the formation of a secondary bud at the base of the progenitor mother cell. Our analyses further revealed there was no correlation between the stage of bud formation by the initial bud (early, mid-late; *Ouellette et al., 2022*), and the appearance of nascent divisomes at the base of the progenitor mother cell (data not shown).

The *Ct* divisome is hybrid in nature, containing elements of the divisome (FtsK and PBP3) and elongasome (PBP2, MreB, and MreC) from other bacteria. Each of these proteins formed foci at the septum, foci at the septum and at the base of the mother cell, or foci only at the base of the mother cell, and the foci of each protein were restricted to one side of the MOMP-stained septum (*Figures 1 and 2*). Knockdown of *ftsK* using CRISPRi revealed that FtsK is necessary for the assembly of this hybrid divisome complex (*Figure 5*). Knockdown and inhibitor studies further revealed that PBP2 is necessary for the addition of PBP3 to the divisome in *Ct* (*Figures 5 and 6*).

The chlamydial divisome proteins all form foci in coccoid cells (*Figures 1 and 2*), and FtsK forms foci in dividing *Ct* that only partially overlap the distribution of PBP2, PBP3, MreC, and MreB (*Figure 2*, *Figure 2—figure supplement 4A*). The lack of overlap of FtsK with MreB and MreC is most clearly evident in *Figure 2—figure supplement 4A*, and while it is unclear why FtsK only partially overlaps the distribution of the other divisome components in dividing *Ct*, MreB (*Liechti et al., 2014*; *Kemege et al., 2015*; *Lee et al., 2020*) and MreC (*Figure 2—figure supplement 5A*) can reorganize into rings in dividing cells, and the MreC rings we detected, like PG rings (*Figure 7*), were at an angle relative to the MOMP-stained septum. MreC also forms rings in coccoid cells (*Figure 2—figure supplement 5B*). Although MreB and MreC form rings in *Ct*, we never detected FtsK rings in dividing or coccoid chlamydial cells. The relevance of the angled orientation of PG and MreC rings relative to the MOMP-stained septum in division intermediates is unclear. However, it appears to be a conserved feature of the cell division process and may arise because the divisome proteins are often positioned slightly above or below the plane of the MOMP-stained septum (*Figures 1 and 2*).

Previous studies hypothesized that MreB filaments may substitute for FtsZ and form a scaffold necessary for the assembly of the divisome in *Ct* (*Ouellette et al., 2012*; *Ouellette et al., 2015*; *Ouellette et al., 2020*). However, our analyses have indicated that MreB is one of the last components recruited to nascent divisomes that form at the base of the mother cell in *Ct*, and localization studies revealed that foci formation by FtsK, PBP2, and PBP3 are not dependent on MreB filament formation (*Figure 3*). Although our data indicate that MreB filaments do not form a scaffold necessary for the assembly of all components of the divisome in *Ct*, MreB filaments are necessary for the conversion of PG foci into PG rings in *Ct* (*Figure 7*).

FtsZ treadmilling drives its rotational movement at the septum and this may be required for the positioning of peptidoglycan biosynthetic enzymes at the division plane in gram-negative and gram-positive bacteria (*Bisson-Filho et al., 2017*; *Yang et al., 2017*). The knockdown studies presented here demonstrated that in the absence of FtsZ, chlamydial FtsK is critical for divisome assembly and PG synthesis in *Ct*, and future real-time imaging studies will determine whether changes in the organization/distribution of FtsK occur during chromosomal translocation and/or cell division in *Chlamydia*. Furthermore, future studies will investigate the mechanisms that regulate the site of assembly of nascent divisomes in the mother cell during the polarized cell division process.

*Ct* is a member of the Planctomycetes/Verrucomicrobia/Chlamydia superphylum and members of the Chlamydia and Planctomycetes phyla do not encode FtsZ (*Rivas-Marín et al., 2016*). *Planctospirus limnophila* is a member of the Planctomycetes that divides by polarized budding, and recent knockout studies (*Rivas-Marin et al., 2023*) indicated that FtsK is the only protein of the chlamydial divisome we characterized here that is essential for the growth of this free-living organism. These results suggest that multiple mechanisms of FtsZ-independent polarized budding have evolved in members of this superphyla. It will be of interest in future studies to determine whether other members of the Planctomycetes that bud (*Wiegand et al., 2020*) divide using a divisome apparatus similar to *Ct*.

## Materials and methods
### Cell culture
HeLa cells (clone S3, authenticated by ATCC, Manassas, VA) were cultured in Dulbecco's Modified Eagle Medium (DMEM; Invitrogen, Waltham, MA) containing 10% fetal bovine serum (FBS, Hyclone,

Logan, UT) at 37 °C in a humidified chamber with 5% CO2. HeLa were only passaged five times before fresh aliquots of cells were thawed for use. The Hela cell stock was routinely tested for mycobacterial contamination by PCR. HeLa cells grown in DMEM containing 10% FBS were infected with *Ct* serovar L2 434/Bu. Infections of HeLa cells with chlamydial transformants were performed in DMEM containing 10% FBS and 0.36 U/mL penicillin G (Sigma-Aldrich).

## Cloning

The plasmids and primers used for generating mCherry fusions of FtsK, PBP2, PBP3, and MreC are listed in Supp. File 1 A. The chlamydial *ftsK, pbp2, pbp3,* and *mreC* genes were amplified by PCR with Phusion DNA polymerase (NEB, Ipswich, MA) using 10 ng *C. trachomatis* serovar L2 genomic DNA as a template. The PCR products were purified using a PCR purification kit (Qiagen) and inserted into the pBOMB4-Tet (-GFP) plasmid, which confers resistance to β-lactam antibiotics. The plasmid was cut at the NotI (FtsK-mCherry) or the KpnI (mCherry-PBP2, mCherry-PBP3, mCherry-MreC) site, and the chlamydial genes were inserted into the cut plasmid using the HiFi DNA Assembly kit (NEB) according to the manufacturer's instructions. The products of the HiFi reaction were transformed into NEB-5αI[q] competent cells (NEB) and transformants were selected by growth on plates containing ampicillin. DNA from individual colonies was isolated using a mini-prep DNA isolation kit (Qiagen), and plasmids were initially characterized by restriction digestion to verify the inserts were the correct size. Clones containing inserts of the correct size were DNA sequenced prior to use.

## DNA and RNA purification and RT-qPCR

Total nucleic acids were extracted from HeLa cells infected with *Ct* plated in 6-well dishes as described previously (*Ouellette et al., 2015*; *Ouellette et al., 2021*). For RNA isolation, cells were rinsed one time with PBS, then lysed with 1 mL Trizol (Invitrogen) per well. Total RNA was extracted from the aqueous layer after mixing with 200 µL per sample of chloroform following the manufacturer's instructions. Total RNA was precipitated with isopropanol and treated with DNase (Ambion) according to the manufacturer's guidelines prior to cDNA synthesis using SuperScript III (Invitrogen). For DNA, infected cells were rinsed one time with PBS, trypsinized and pelleted before resuspending each pellet in 500 µL of PBS. Each sample was split in half, and genomic DNA was isolated from each duplicate sample using the DNeasy extraction kit (Qiagen) according to the manufacturer's guidelines. Quantitative PCR was used to measure *C. trachomatis* genomic DNA (gDNA) levels using an *euo* primer set. 150 ng of each sample was used in 25 µL reactions using standard amplification cycles on a Quant-Studio3 thermal cycler (Applied Biosystems) followed by a melting curve analysis. *ftsK, pbp2, euo*, and *omcB* transcript levels were determined by RT-qPCR using SYBR Green as described previously (*Ouellette et al., 2021*) (see Supp. File 1B for primers used for measuring gDNA levels and RT-qPCR). Transcript levels were normalized to genomes and expressed as ng cDNA/gDNA.

## Transformation of *Ct*

*Ct* was transformed as described previously (*Wang et al., 2011*). Briefly, HeLa cells were plated in a 10 cm plate at a density of $5 \times 10^6$ cells the day before beginning the transformation procedure. *Ct* lacking its endogenous plasmid (-pL2) was incubated with 10 µg of plasmid DNA in Tris-CaCl$_2$ buffer (10 mM Tris-Cl pH 7.5, 50 mM CaCl$_2$) for 30 min at room temperature. HeLa cells were trypsinized, washed with 8 mL of 1 x DPBS (Gibco), and pelleted. The pellet was resuspended in 300 µL of the Tris-CaCl$_2$ buffer. *Ct* was mixed with the HeLa cells and incubated at room temperature for an additional 20 min. The mixture was added to 10 mL of DMEM containing 10% FBS and 10 µg/mL gentamicin and transferred to a 10 cm plate. At 48 hpi, the HeLa cells were harvested and *Ct* in the population were used to infect a new HeLa cell monolayer in media containing 0.36 U/ml of penicillin G to select for transformants. The plate was incubated at 37 °C for 48 hr. These harvest and re-infection steps were repeated every 48 hr until inclusions were observed.

## Immunofluorescence microscopy

HeLa cells were seeded in 10 cm plates at a density of $5 \times 10^6$ cells per well the day before the infection. *Ct* L2 or chlamydial strains transformed with plasmids encoding FtsK-mCherry, mCherry-PBP2, mCherry-PBP3, or mCherry-MreC or with plasmids that direct the constitutive expression of the crRNAs targeting the *pbp2* or *ftsK* promoters were used to infect HeLa cells from ATCC?? in DMEM.

For experiments with the transformants, aTc was added to the media of infected cells at the indicated concentration and time. At 21 hpi, cells were detached from the 10 cm plate by scraping and pelleted by centrifugation for 30 s. The pellet was resuspended in 1 mL of 0.1 x PBS (Gibco) and transferred to a 2 mL tube containing 0.5 mm glass beads (Thermo Fisher Scientific). Cells were vortexed for 3 min. then centrifuged at 800 rpm for 2 min in a microfuge. 20 µLs of the supernatant was mixed with 20 µLs of 2 x fixing solution (6.4% formaldehyde and 0.044% glutaraldehyde) and incubated on a glass slide for 10 min at room temperature. Cells were washed with three times with PBS, and the cells were permeabilized by incubation with PBS containing 0.1% Triton X-100 for 1 min. Cells were washed with PBS two times. For experiments with *Ct* L2, the cells were incubated with a goat primary antibody against the major outer-membrane protein (MOMP; Meridian, Memphis, TN), and the mouse primary antibody that recognizes endogenous FtsK raised against recombinant CT739 protein (https://doi.org/10.1099/mic.0.047746-0), or with rabbit antibodies raised against peptides derived from PBP 2 or PBP3 (*Ouellette et al., 2012*). Briefly, chlamydial antigens or peptides emulsified with Freund's incomplete adjuvant were used to immunize animals via intramuscular injections three times with an interval of 2 wk. Antisera were collected from the immunized animals 2–4 wk after the final immunization as the primary antibodies. After the primary antibody labeling, the cells were then rinsed with PBS and incubated with donkey anti-goat IgG (Alexa 488) and donkey anti-mouse IgG (Alexa 594) or donkey-anti-rabbit IgG (Alexa 594) secondary antibodies (Invitrogen). Experiments in which we visualized the distribution of the various mCherry fusions, the localization of the mCherry fluorescence was compared to the distribution of MOMP. In some experiments, we determined the distribution of the MreB_6 x His fusion by staining cells expressing the fusion with a rabbit anti-6x His antibody (Abcam, Cambridge, MA) and the goat anti-MOMP antibody followed by the appropriate secondary antibodies. Cells were imaged using Zeiss AxioImager2 microscope equipped with a 100 x oil immersion PlanApochromat objective and a CCD camera. During image acquisition, 0.3 µm xy-slices were collected that extended above and below the cell. The images were collected such that the brightest spot in the image was saturated. The images were deconvolved using the nearest neighbor algorithm in the Zeiss Axiovision 4.7 software. Deconvolved images were viewed and assembled using Zeiss Zen-Blue software. For each experiment, three independent replicates were performed, and the values shown for localization are the average of the three experiments. In some instances, 3D projections of the acquired xy slices were generated using the Zeiss Zen-Blue software.

## Peptidoglycan (PG) labeling

PG was labeled by incubating cells with 4 mM ethylene-D-alanine-D-alanine (E-DA-DA) as described (*Cox et al., 2020*). The incorporated E-DA-DA was fluorescently labeled using the Click & Go labeling kit (Vector Laboratories). The distribution of fluorescently labeled PG was compared to the distribution of MOMP and endogenous FtsK or the distribution of mCherry-PBP3. Three independent replicates were performed, and the values shown are the average of the three experiments.

## Inclusion forming unit assay

HeLa cells were infected with *Ct* (-pL2) transformed with the pBOMB4 Tet (-GFP) plasmid encoding the indicated aTc-inducible gene. At 8 hpi, aTc was added to the culture media at the indicated concentration. Control cells were not induced. At 48 hpi, the HeLa cells were dislodged from the culture dishes by scraping and collected by centrifugation. The pellet was resuspended in 1 mL of 0.1 x PBS (Gibco) and transferred to a 2 mL tube containing 0.5 mm glass bead tubes (Thermo Fisher Scientific). Cells were vortexed for 3 min followed by centrifugation at 800 rpm for 2 min. The supernatants were mixed with an equal volume of a 2 x sucrose-phosphate (2SP) solution (ref) and frozen at –80 °C. At the time of the secondary infection, the chlamydiae were thawed on ice and vortexed. Cell debris was pelleted by centrifugation for 5 min at 1 k x g at 4 °C. The EBs in the resulting supernatant were serially diluted and used to infect a monolayer of HeLa cells in a 24-well plate. The secondary infections were allowed to grow at 37 °C for 24 hr before they were fixed and labeled for immunofluorescence microscopy by incubating with a goat anti-MOMP antibody followed by a secondary donkey anti-goat antibody (Alexa Fluor 594). The cells were rinsed in PBS and inclusions were imaged using an EVOS imaging system (Invitrogen). The number of inclusions were counted in five fields of view and averaged. Three independent replicates were performed, and the values from the replicates

were averaged to determine the number of inclusion-forming units. Chi-squared analysis was used to compare IFUs in induced and uninduced samples.

## Effect of A22 and mecillinam on the profile of division intermediates and on PG and divisome protein localization in *Ct*

HeLa cells were infected with *Ct* transformed with the pBOMB4-Tet (-GFP) plasmid encoding FtsK-mCherry, mCherry-PBP2, mCherry-PBP3, mCherry-MreC, or MreB-6xHis. The fusions were induced at 20 hpi with 10 nM aTc for 1 hr in the absence or presence of 75 µM A22. At 22 hpi, cells were harvested and prepared for staining as described above. Three independent replicates were performed, and the values shown for localization are the average of the three experiments.

HeLa cells were infected with *Ct* L2 and 20 µM mecillinam (Sigma) was added to the media of infected cells at 17 hpi. Control cells were untreated. At 22 hpi, infected cells were harvested and RBs were prepared and stained with MOMP, FtsK, PBP2, or PBP3 antibodies as described above. Alternatively, cells were incubated with 4 mM EDA-DA at 17hpi in the presence or absence of 20 µM mecillinam. The cells were harvested at 22 hpi, and RBs were prepared and PG was click-labeled and its distribution was visualized in MOMP-stained cells as described above. Three independent replicates were performed, and the values shown for localization are the average of the three experiments.

### Immunoblotting

HeLa cells infected with *Ct* L2 were harvested by scraping the infected cells from the plate at 24 hpi. Uninfected HeLa cells were included as a control. The HeLa cells were pelleted by centrifugation, resuspended in SDS sample buffer, and electrophoresed on a 10% SDS polyacrylamide gel. The gel was electrophoretically transferred to nitrocellulose (Schleicher and Schuell), and the filter was incubated with mouse polyclonal antibodies raised against chlamydial FtsK. The filter was rinsed and incubated with 800 donkey anti-mouse IgG secondary antibodies (LICOR, Lincoln, NE) and imaged using a LICOR Odyssey imaging system.

HeLa cells were infected with *Ct* transformed with plasmids that inducibly express FtsK-mCherry, mCherry-PBP2, mCherry-PBP3, or mCherry-MreC. The fusions were induced by the addition of 10 nM aTc to the media of infected cells at 17 hpi. The cells were harvested and pelleted at 21 hpi. The cell pellet was resuspended in 1 mL of 0.1 x PBS (Gibco) and transferred to a 2 mL tube containing 0.5 mm glass beads (Thermo Fisher Scientific). Cells were vortexed for 3 min followed by centrifugation at 800 rpm for 2 min. The supernatant was collected and centrifuged for 3 min at 13,000 rpm and the pellet containing *Ct* was resuspended in TBS containing 1% TX-100, 1 X protease inhibitor cocktail (Sigma), and 5 µM lactacystin. The suspension was sonicated three times on ice and centrifuged at 13,000 rpm for 3 min. The supernatant was collected and mixed with SDS sample buffer. The samples were boiled and electrophoresed on a 10% SDS polyacrylamide gel, and the gel was electrophoretically transferred to nitrocellulose. The blots from these analyses were probed with a rabbit anti-mCherry primary antibody (Invitrogen) and a 800 donkey anti-rabbit IgG secondary antibodies (LICOR, Lincoln, NE). The filters were imaged using a LICOR Odyssey imaging system.

HeLa cells were infected with *Ct* transformed with the pBOMB4-Tet (-GFP) plasmid encoding mCherry-PBP2 or mCherry-PBP3. The fusions were induced with 10 nM aTc at 17 hpi. Uninduced cells were included as a control. The cells were harvested at 21 hpi, and samples were processed for immunoblotting as described above. The blots were probed with rabbit polyclonal antibodies raised against peptides derived from chlamydial PBP2 or PBP3 (*Ouellette et al., 2012*). The blots were then rinsed and incubated with 800 donkey anti-rabbit IgG secondary antibodies (LICOR, Lincoln, NE). The filters were imaged using a LICOR Odyssey imaging system.

## Acknowledgements

We thank Dr. I Clarke (University of Southampton) for providing the plasmidless strain of *C. trachomatis* serovar L2.

## Additional information

### Funding

| Funder | Grant reference number | Author |
|--------|------------------------|--------|
| National Institute of General Medical Sciences | R35GM151971 | Scot P Ouellette |
| National Science Foundation | 1817578 | John V Cox |

The funders had no role in study design, data collection and interpretation, or the decision to submit the work for publication.

### Author contributions

McKenna Harpring, Conceptualization, Formal analysis, Methodology, Writing – original draft; Jung-hoon Lee, Formal analysis, Methodology; Guangming Zhong, Resources; Scot P Ouellette, Conceptualization, Formal analysis, Supervision, Funding acquisition, Methodology, Writing – review and editing; John V Cox, Conceptualization, Formal analysis, Supervision, Funding acquisition, Methodology, Writing – original draft, Project administration

### Author ORCIDs

McKenna Harpring ⬛ https://orcid.org/0009-0007-4178-244X
Scot P Ouellette ⬛ https://orcid.org/0000-0002-3721-6839
John V Cox ⬛ https://orcid.org/0000-0002-6177-0223

Reviewer #1 (Public review): https://doi.org/10.7554/eLife.104199.3.sa1
Reviewer #2 (Public review): https://doi.org/10.7554/eLife.104199.3.sa2
Reviewer #3 (Public review): https://doi.org/10.7554/eLife.104199.3.sa3
Author response https://doi.org/10.7554/eLife.104199.3.sa4

---

## Additional files

### Supplementary files

Supplementary file 1. Primers and plasmids used in our analyses. (A) List of primers and plasmids used for cloning mCherry fusions of FtsK, PBP2, PBP3, MreB, and MreC. (B) List of primers used for RT-qPCR.

MDAR checklist

### Data availability

Original source data for blots and microscopy images have been uploaded as three DOIs to Dryad. Source data for microscopy images and immunoblots are available at Dryad (Figures 1–7 IF images - https://doi.org/10.5061/dryad.4qrfj6qnh; gels/blots - https://doi.org/10.5061/dryad.1ns1rn94w; supplemental IF images - https://doi.org/10.5061/dryad.2bvq83c27).

The following datasets were generated:

| Author(s) | Year | Dataset title | Dataset URL | Database and Identifier |
|-----------|------|---------------|-------------|-------------------------|
| Harpring M, Cox JV, Lee J, Ouellette S, Zhong G | 2025 | Source data: FtsK is Critical for the Assembly of the Unique Divisome Complex of the FtsZ-less *Chlamydia trachomatis* Figures 1-7 | https://doi.org/10.5061/dryad.4qrfj6qnh | Dryad Digital Repository, 10.5061/dryad.4qrfj6qnh |

*Continued on next page*

*Continued*

| Author(s) | Year | Dataset title | Dataset URL | Database and Identifier |
|---|---|---|---|---|
| Harpring M, Cox JV, Lee J, Ouellette S, Zhong G | 2025 | FtsK is Critical for the Assembly of the Unique Divisome Complex of the FtsZ-less *Chlamydia trachomatis* Blots | https://doi.org/10.5061/dryad.1ns1rn94w | Dryad Digital Repository, 10.5061/dryad.1ns1rn94w |
| Harpring M, Cox JV, Ouellette S, Zhong G, Lee J | 2025 | Source data: FtsK is Critical for the Assembly of the Unique Divisome Complex of the FtsZ-less *Chlamydia trachomatis* IF Images Supplemental Figures | https://doi.org/10.5061/dryad.2bvq83c27 | Dryad Digital Repository, 10.5061/dryad.2bvq83c27 |

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
