## [Editor Report · eLife Assessment]

Understanding how the divisome is assembled in Chlamydia trachomatis, a bacterial pathogen, is crucial since this bacterium has a non-canonical cell wall and lacks the master regulator of cell division, FtsZ. This **important** study shows that a DNA translocase, FtsK, is an early and essential component of the *Chlamydia trachomatis* divisome. The evidence presented is **convincing**, leveraging the elegant use of genetics and fluorescence microscopy. As this role of FtsK is distinct relative to most other bacteria, these findings should be of significant interest to bacterial cell biologists.

---

## [Referee Report · Reviewer #1 (Public review)]

Summary:

In this work, Harpring et al. investigated divisome assembly in *Chlamydia trachomatis* serovar L2 (Ct), an obligate intracellular bacterium that lacks FtsZ, the canonical master regulator of bacterial cell division. They find that divisome assembly is initiated by the protein FtsK in Ct by showing that it forms discrete foci at the septum and future division sites. Additionally, knocking down ftsK prevents divisome assembly and inhibits cell division, further supporting their hypothesis that FtsK regulates divisome assembly. Finally, they show that MreB is one of the last chlamydial divisome proteins to arrive at the site of division and is necessary for the formation of septal peptidoglycan rings but does not act as a scaffold for division assembly as previously proposed.

Strengths:

The authors use microscopy to clearly show that FtsK forms foci both at the septum as well as at the base of the progenitor cell where the next septum will form. They also show that the Ct proteins PBP2, PBP3, MreC, and MreB localize to these same sites suggesting they are involved in the divisome complex.

Using CRISPRi the authors knockdown ftsK and find that most cells are no longer able to divide and that PBP2 and PBP3 no longer localized to sites of division suggesting that FtsK is responsible for initiating divisome assembly. They also performed a knockdown of pbp2 using the same approach and found that this also mostly inhibited cell division. Additionally, FtsK was still able to localize in this strain however PBP3 did not suggest that FtsK acts upstream of PBP2 in the divisome assembly process while PBP2 is responsible for the localization of PBP3.

The authors also find that performing a knockdown of ftsK also prevents new PG synthesis further supporting the idea that FtsK regulates divisome assembly. They also find that inhibiting MreB filament formation using A22 results in diffuse PG, suggesting that MreB filament formation is necessary for proper PG synthesis to drive cell division.

Overall the authors propose a new hypothesis for divisome assembly in an organism that lacks FtsZ and use a combination of microscopy and genetics to support their model that is rigorous and convincing. The finding that FtsK, rather than a cytoskeletal or "scaffolding" protein is the first division protein to localize to the incipient division site is unexpected and opens up a host of questions about its regulation. The findings will progress our understanding of how cell division is accomplished in bacteria with non-canonical cell wall structure and/or that lack FtsZ.

Weaknesses:

No major weaknesses were noted in the data supporting the main conclusions.

---

## [Referee Report · Reviewer #2 (Public review)]

Summary:

Chlamydial cell division is a peculiar event, whose mechanism was mysterious for many years. *C. trachomatis* division was shown to be polar and involve a minimal divisome machinery composed of both homologues of divisome and elongasome components, in absence of an homologue of the classical division organizer FtsZ. In this paper, Harpring et al., show that FtsK is required at an early stage of the chlamydial divisome formation.

Strengths:

The manuscript is well-written and the results are convincing. Quantification of divisome component localization is well performed, number of replicas and number of cells assessed are sufficient to get convincing data. The use of a CRISPRi approach to knock down some divisome components is an asset and allows a mechanistic understanding of the hierarchy of divisome components

Weaknesses

Despite advances in the understanding of the importance of FtsK for chlamydial division, this manuscript does not show by which mechanism FtsK specifically localizes at the division site and how it mediates recruitment of other divisome members. Moreover, the potential link with DNA partitioning is not addressed.

---

## [Referee Report · Reviewer #3 (Public review)]

Summary:

The obligate intracellular bacterium *Chlamydia trachomatis* (Ct) divides by binary fission. It lacks FtsZ, but still has many other proteins that regulate synthesis of septal peptidoglycan, including FtsW and FtsI (PBP3) as well as divisome proteins that recruit and activate them, such as FtsK and FtsQLB. Interestingly, MreB is also required for division of Ct cells, perhaps by polymerizing to form an FtsZ-like scaffold. Here, Harpring et al. show that MreB does not act early in division and instead is recruited to a protein complex that includes FtsK and PBP2/PBP3. This indicates that Ct cell division is organized by a chimera between conserved divisome and elongasome proteins. Their work also shows convincingly that FtsK is the earliest known step of divisome activity, potentially nucleating the divisome as a single protein complex at the future division site. This is reminiscent of the activity of FtsZ, yet fundamentally different.

Strengths:

The study is very well written and presented, and the data are convincing and rigorous. The data underlying the proposed localization dependency order of the various proteins for cell division is well justified by several different approaches using small molecule inhibitors, knockdowns, and fluorescent protein fusions. The proposed dependency pathway of divisome assembly is consistent with the data and with a novel mechanism for MreB in septum synthesis in Ct.

Weaknesses:

The authors have addressed the weaknesses brought up in my previous review.

---

## [Author Response]

The following is the authors’ response to the original reviews.

**Public Reviews:**

**Reviewer #1 (Public review):**
Summary:In this work, Harpring et al. investigated divisome assembly in *Chlamydia trachomatis* serovar L2 (Ct), an obligate intracellular bacterium that lacks FtsZ, the canonical master regulator of bacterial cell division. They find that divisome assembly is initiated by the protein FtsK in Ct by showing that it forms discrete foci at the septum and future division sites. Additionally, knocking down ftsK prevents divisome assembly and inhibits cell division, further supporting their hypothesis that FtsK regulates divisome assembly. Finally, they show that MreB is one of the last chlamydial divisome proteins to arrive at the site of division and is necessary for the formation of septal peptidoglycan rings but does not act as a scaffold for division assembly as previously proposed.Strengths:The authors use microscopy to clearly show that FtsK forms foci both at the septum as well as at the base of the progenitor cell where the next septum will form. They also show that the Ct proteins PBP2, PBP3, MreC, and MreB localize to these same sites suggesting they are involved in the divisome complex.Using CRISPRi the authors knock down ftsK and find that most cells are no longer able to divide and that PBP2 and PBP3 no longer localized to sites of division suggesting that FtsK is responsible for initiating divisome assembly. They also performed a knockdown of pbp2 using the same approach and found that this also mostly inhibited cell division. Additionally, FtsK was still able to localize in this strain, however PBP3 did not, suggesting that FtsK acts upstream of PBP2 in the divisome assembly process while PBP2 is responsible for the localization of PBP3.The authors also find that performing a knockdown of ftsK also prevents new PG synthesis further supporting the idea that FtsK regulates divisome assembly. They also find that inhibiting MreB filament formation using A22 results in diffuse PG, suggesting that MreB filament formation is necessary for proper PG synthesis to drive cell division.Overall the authors propose a new hypothesis for divisome assembly in an organism that lacks FtsZ and use a combination of microscopy and genetics to support their model that is rigorous and convincing. The finding that FtsK, rather than a cytoskeletal or "scaffolding" protein is the first division protein to localize to the incipient division site is unexpected and opens up a host of questions about its regulation. The findings will progress our understanding of how cell division is accomplished in bacteria with non-canonical cell wall structure and/or that lack FtsZ.Weaknesses:No major weaknesses were noted in the data supporting the main conclusions. However, there was a claim of novelty in showing that multiple divisome complexes can drive cell wall synthesis simultaneously that was not well-supported (i.e. this has been shown previously in other organisms). In addition, there were minor weaknesses in data presentation that do not substantially impact interpretation (e.g. presenting the number of cells rather than the percentage of the population when quantifying phenotypes and showing partial western blots instead of total western blots).

We agree with the weaknesses identified by the reviewer. We removed the statements in the Results and Discussion that multiple independent divisome complexes can simultaneously direct PG synthesis. We presented the data in Figs. 3-5 as % of the cells in the population, and complete western blots are shown in Supp. Fig. S1.

**Reviewer #2 (Public review):**
Summary:Chlamydial cell division is a peculiar event, whose mechanism was mysterious for many years. *C. trachomatis* division was shown to be polar and involve a minimal divisome machinery composed of both homologues of divisome and elongasome components, in the absence of an homologue of the classical division organizer FtsZ. In this paper, Harpring et al., show that FtsK is required at an early stage of the chlamydial divisome formation.Strengths:The manuscript is well-written and the results are convincing. Quantification of divisome component localization is well performed, number of replicas and number of cells assessed are sufficient to get convincing data. The use of a CRISPRi approach to knock down some divisome components is an asset and allows a mechanistic understanding of the hierarchy of divisome components.Weaknesses:The authors did not analyse the role of all potential chlamydial divisome components and did not show how FtsK may initiate the positioning of the divisome. Their conclusion that FtsK initiates the assembly of the divisome is an overinterpretation and is not backed by the data. However, data show convincingly that FtsK, if perhaps not the initiator of chlamydial division, is definitely an early and essential component of the chlamydial divisome.

The following statement has been included in the Discussion (pg. 16 of the revised manuscript) “Although we focused our study on a subset of the divisome and elongasome proteins that *Chlamydia* expresses (bolded in Fig. 6G), our results support our conclusion that chlamydial budding is dependent upon a hybrid divisome complex and that FtsK is required for the assembly of this hybrid divisome. At this time, we cannot rule out that other proteins act upstream of FtsK to initiate divisome assembly in this obligate intracellular bacterial pathogen.”

We will soon be submitting another manuscript that addresses how FtsK specifies the site of divisome assembly. This work is too extensive to be included in this manuscript.

**Reviewer #3 (Public review):**
Summary:The obligate intracellular bacterium *Chlamydia trachomatis* (Ct) divides by binary fission. It lacks FtsZ, but still has many other proteins that regulate the synthesis of septal peptidoglycan, including FtsW and FtsI (PBP3) as well as divisome proteins that recruit and activate them, such as FtsK and FtsQLB. Interestingly, MreB is also required for the division of Ct cells, perhaps by polymerizing to form an FtsZ-like scaffold. Here, Harpring et al. show that MreB does not act early in division and instead is recruited to a protein complex that includes FtsK and PBP2/PBP3. This indicates that Ct cell division is organized by a chimera between conserved divisome and elongasome proteins. Their work also shows convincingly that FtsK is the earliest known step of divisome activity, potentially nucleating the divisome as a single protein complex at the future division site. This is reminiscent of the activity of FtsZ, yet fundamentally different.Strengths:The study is very well written and presented, and the data are convincing and rigorous. The data underlying the proposed localization dependency order of the various proteins for cell division is well justified by several different approaches using small molecule inhibitors, knockdowns, and fluorescent protein fusions. The proposed dependency pathway of divisome assembly is consistent with the data and with a novel mechanism for MreB in septum synthesis in Ct.Weaknesses:The paper could be improved by including more information about FtsK, the "focus" of this study. For example, if FtsK really is the FtsZ-like nucleator of the Ct divisome, how is the Ct FtsK different sequence-wise or structurally from FtsK of, e.g. *E. coli*? Is the N-terminal part of FtsK sufficient for cell division in Ct like it is in *E. coli*, or is the DNA translocase also involved in focus formation or localization? Addressing those questions would put the proposed initiator role of FtsK in Ct in a better context and make the conclusions more attractive to a wider readership.

We will be submitting another manuscript soon that details the conserved domain organization of FtsK from different bacteria, and the role of the various domains of chlamydial FtsK (including the N-terminus and the C-terminal translocase domain) in directing its localization in dividing *Chlamydia*. We have added text to the discussion (pg. 16 of the revised manuscript) that describes the sequence homology of chlamydial FtsK to FtsK from *E. coli.*

Another weakness is that the title of the paper implies that FtsK alone initiates divisome assembly. However, the data indicate only that FtsK is important at an early stage of divisome assembly, not that it is THE initiator. I suggest modifying the title to account for this--perhaps "FtsK is required to initiate....".

We agree with the reviewer and modified the title to “FtsK is Critical for the Assembly of the Unique Divisome Complex of the FtsZ-less *Chlamydia trachomatis”.* We have also modified the text throughout to indicate that FtsK is required for the assembly of the hybrid divisome of *Chlamydia*.

**Recommendations for the authors:**

**Reviewer #1 (Recommendations for the authors):**
Suggestions for improvement (mostly minor):(1) For several of the graphs, the authors plot the number of cells with a given phenotype on the y-axis, but then describe percentages of cells in the text. It would make it clearer if all the graphs had the percentage of cells on the y-axis instead.

We have modified the figures to indicate the percentage of cells on the y-axis with a given phenotype.

(2) In Figures 3, 4, and 5 the authors show separate graphs for plus/minus drug or inducer. These should be on the same graph as they are directly comparing these two different conditions. Having them on separate graphs makes it less clear whether these differences are significant between the two conditions

We modified Fig. 4 to show +/- inducer in *ftsk* and *pbp2* knockdown strains in the same graph. Regarding Figures 3 and 5, we believe the figures in the original submission effectively demonstrate the +/- drug conditions, so these figures remain unchanged in the revised manuscript.

(3) In Figure 2 the authors show microscopy of the colocalization of FtsK with several other divisome proteins from Ct. Quantification of the colocalization of FtsK with these other proteins would provide a more holistic understanding of their colocalization and help further support their argument that FtsK initiates the assembly of the divisome.

Supp. Fig. S4A of the revised manuscript contains images showing the colocalization of FtsK with the fusions at the septum and the base of dividing cells, and the colocalization of FtsK with the fusions that are only at the base of dividing cells. Supp. Fig. S4B quantified the percentage of dividing cells where FtsK overlaps the localization of each of the fusions at the septum, at the septum and the base, and at the base alone.

(4) In Figure 6 the authors mention that the PG ring was at a slight angle relative to the MOMP-stained septum. What is the significance of this? The authors mention it several times but do not explain its relevance to divisome assembly. It is not really evident in the images presented.

We mention in the discussion pgs. 17-18 of the revised manuscript that “The relevance of the angled orientation of PG and MreC rings relative to the MOMP-stained septum in division intermediates is unclear. However, it appears to be a conserved feature of the cell division process and may arise because the divisome proteins are often positioned slightly above or below the plane of the MOMP-stained septum. The positioning of divisome proteins above or below the septum is indicated in Figs. 1 and 2.

We included cartoons in Fig. 6C of the revised manuscript to assist the reader in visualizing the angled orientation of the PG ring relative to the MOMP-stained septum.

(5) In line 270 the authors claim that "these are the first data in any system to suggest that septal PG synthesis/modification is simultaneously directed by multiple independent divisome complexes." However, their experiments do not demonstrate that multiple divisome complexes are active at the same time. They show that multiple foci of FtsK etc. are present at sites where PG synthesis has occurred, but that does not necessarily mean that each focus/complex was actively synthesizing PG at the same time. Moreover, similar approaches were used to support a claim that septal PG synthesis is directed by multiple discrete divisome complexes previously (e.g. in Figure 1 of Bisson-Filho et al. 2017 (PMID: 28209898) in *Bacillus subtilis* and in Perez et al 2021 (PMID: 33269494) in Streptococcus pneumoniae). This claim is not central to the main conclusions of the study and could just be removed.

This statement has been removed from the Results and the Discussion.

(6) In Figure 6B the authors see three distinct FtsK foci. Why is this the only place in the manuscript where they see three foci? They mentioned previously that they saw foci at the septum and at the base of the progenitor mother cell, but why are there three foci here?

The vast majority of dividing cells displayed one foci at the septum and/or the base. Representative images were chosen that reflected the localization profiles observed in the majority of cells. While we observed cells with multiple foci, as shown in Figure 6C, these cells were relatively rare (~2% of cells for all the divisome proteins in 3 independent experiments). Since the number of cells with multiple foci were relatively rare, we chose to group these cells with the cells that had single foci at the septum, the septum and base, or base alone categories in the quantification shown in Fig. 2C. This is stated in the legend of Fig. 2 of the revised manuscript.

(7) The Discussion section is lacking a couple of things that would put the data in a broader context. Can the authors speculate on how FtsK knows how to find the division site? I.e. what might be upstream of FtsK localization? Additionally, the authors do not talk about the FtsK sequence or domains at any point in the paper. Does Ct FtsK have a similar sequence/structure to FtsKs from other bacteria? Are there any differences in sequence/structure that might tell us about its function in Ct?

We will be submitting another manuscript soon that examines how the site of assembly of the divisome is defined in dividing *Chlamydia*. This manuscript will also define the localization of the different sub-domains of chlamydial FtsK during cell division. For this manuscript, we added a paragraph in the Discussion (pg. 16 of the revised manuscript) that states the domain organization is conserved in FtsK proteins from different bacteria. This paragraph includes information regarding the % sequence identity of the C-terminus and the N-terminus of chlamydial FtsK when compared to *E. coli* FtsK.

(8) For Supplementary Figure S1B-C. The authors should show the full blots rather than just the single band of the protein of interest to show that the antibodies are specific. Additionally, the authors should include a loading control to show that they loaded the same amount of protein for each sample.

We have included the full blots in Supp. Fig. S1 of the revised manuscript. We do not see the need for including a loading control for these blots because we are not making arguments about the relative level of the proteins that were assayed. We only use the blots to show that the fusion proteins are primarily a single species of the predicted molecular mass.

(9) In Supplementary Figure S4A the authors use RT-qPCR to measure ftsK and pbp2 transcript levels. Since they have antibodies against these proteins, they should also include Western blots to show that the proteins are not being produced when targeted using CRISPRi.

We have included data in Supp. Fig. S5E of the resubmission that indicates foci of FtsK and PBP2 could not be detected following the knockdown of *ftsk* and *pbp2*. We feel that these data support our conclusion that the induced expression of dCas12 in the the *ftsk* and *pbp2* knockdown strains results in the downregulation of the endogenous FtsK and PBP2 polypeptides.

(10) In lines 261-262 the authors say that "PG organization was the same or differed at the septum." What is the PG organization being compared to? Same or different from what?

We agree with the reviewer that the text in lines 261-262 in the original submission was confusing. The text has been modified.

(11) Lines 201-215 the authors refer to Supplementary Figure S3 throughout this section, but they should refer to Supplementary Figure S4.

This has been corrected.

**Reviewer #2 (Recommendations for the authors):**
I am not convinced that this paper shows that FtsK initiates the assembly of the divisome since the authors did not analyse the role and localization of all other chlamydial divisome components. Out of the ten homologues of divisome and elongasome components encoded by *C. trachomatis* genome, only five are investigated in this study. There is no explanation about how these five were chosen.

We state on pg. 16 of the revised manuscript that “Although we focused our study on a subset of the divisome and elongasome proteins that *Chlamydia* expresses (bolded in Fig. 6G), our results support our conclusion that chlamydial budding is dependent upon a hybrid divisome complex and that FtsK is required for the assembly of this hybrid divisome. At this time, we cannot rule out that other proteins act upstream of FtsK to initiate divisome assembly in this obligate intracellular bacterial pathogen.

Results convincingly indicate that FtsK is an early divisome component, but proofs are lacking to indicate that it initiates the divisome formation. Indeed, the authors do not show how FtsK would be the first protein to selectively accumulate at a given location to initiate the divisome formation. For this reason, the model they propose at the end of their study is not backed by sufficient data, to my opinion.

We agree with the reviewer that our data does not show that FtsK initiates divisome assembly. The title of the manuscript has been modified to “FtsK is Critical for the Assembly of the Unique Divisome Complex of the FtsZ-less *Chlamydia trachomatis*” and the text throughout has been modified to indicate that FtsK is the first protein we assayed that associates with nascent divisomes at the base of dividing cells. We will soon be submitting another manuscript that details how FtsK is recruited to a specific site to initiate nascent divisome assembly, This work is too extensive to be included in this manuscript.

There are also discrepancies in the number of cells analysed to quantify the localization of divisome components, ranging from 50 to 250 cells. The authors could better explain why there are such variations.

There were differences in the number of cells analyzed in the various experiments, but in every instance the effect of inhibitors (A22 and mecillinam) or *ftsk* and *pbp2* knockdown on divisome assembly was statistically significant.

There are a few mistakes in the text regarding figure numbering (Figure S4 is mentioned as S3 in the text). Figures 5B and D are not specifically cited.

These mistakes have been corrected in the revised manuscript.

Line 261-262: the sentence starting "Our imaging analysis.." is not clear to me.

We agree with the reviewer that the text in lines 261-262 was confusing. The text has been modified (pg. 14 of the revised manuscript).

Line 270-271: there are insufficient proofs to say that there are multiple independent divisome complexes. This is in my opinion an overinterpretation of the data, since there is no proof that these complexes are independent.

This statement has been removed from the text.

A few details are lacking in the figure legends:Figure 2C: when was the expression of the different mCherry and 6xHis constructs induced?

The onset and length of the induction of the fusions have been included in the legend of Fig. 2.

Bars are sometimes mentioned as uM and should be um. Bars sizes, number of replicates, and/or meaning of the error bars are lacking in legends of Figures S2, S3, and S4

This has been corrected in the revised manuscript.

The consistency of Figures could be improved between Figures 3A, 4A, B, and 5A. The results of treated cells could be always shown as dark grey. It would help the reader.

We have used consistent coloring in Figs. 3-5 to indicate the treated cells.

**Reviewer #3 (Recommendations for the authors):**
(1) Lines 113-118: do Ct cells increase in size as they get closer to starting division? If so, could a pseudo-time course (demograph) be done to bolster the evidence that the base foci formed mainly in predivisional cells and not newborn cells? This evidence might be more convincing than the data in Figures 1F and G.

Chlamydial cells in the population were heterogeneous in size at the timepoint we are studying. This observation is consistent with previous reports in the literature (Liechti et al.,2021). While we agree that a pseudo-time course could potentially bolster the evidence about when FtsK foci appear, we believe our current analysis sufficiently demonstrates that basal foci of FtsK appear prior to the appearance of new buds at the base of dividing cells.

(2) Figure 3E: It looks like MreC localization to foci doesn't strictly require MreB polymerization. Is this known for *E. coli* or other species?

To our knowledge, MreC assembly into a filament has not been shown to be dependent upon MreB in other bacteria. In *Caulobacter crescentus*, MreC forms a helical structure that is not dependent upon MreB or MreB filament formation (Dye et al., 2005. PNAS; Divakaruni et al., 2005. PNAS).

(3) Figure 5E: why is nearly half of PBP2 and PBP3 still localized to foci at the membrane even after treatment with mecillinam? This suggests, as the authors mention, that mecillinam reduces the efficiency of localization to the divisome but does not eliminate it. Any ideas why?

At this time, we do not know why inhibiting the catalytic activity of PBP2 with mecillinam does not fully prevent the association of PBP2 with the chlamydial divisome. We have included a statement in the Results (pg. 13 of the revised manuscript) that inhibiting the catalytic activity of PBP2 prevents it from efficiently associating with or maintaining its association with polarized divisome complexes.

(4) Line 262-263: This sentence is confusing-please rephrase. The same as what? Differed from what?

We agree with the reviewer. The wording in lines 262-263 of the original submission has been modified.

(5) Lines 265-267 and Figure 6: Adding cartoon schematics might help readers visualize cell orientations in Fig. 6 (especially 6B).

Cartoons have been added to Fig. 6C (Fig. 6B in the original submission) to orient the reader.

(6) Line 294-298: Do the authors think that the residual 5-10% of PG foci after FtsK knockdown is due to the ability of residual FtsK to organize divisomes?

We show that knockdown of FtsK is not complete, and while we cannot be certain, it is likely, that the PG foci detected in FtsK knockdown cells is due to the ability of the residual FtsK to organize divisomes that direct PG synthesis.

(7) Do the authors have any evidence that FtsK foci are mobile like treadmilling FtsZ?

We have not performed real-time imaging studies, and we currently have no evidence that FtsK foci are mobile.

(8) FtsK foci here are reminiscent of mobile foci formed by the FtsK-like SpoIIIE at the *Bacillus subtilis* sporulation septum. This might be a good idea to mention in the Discussion. Is it possible that Ct FtsK is also involved in coordinating chromosome partitioning through the developing septum? (That is another reason why it would be useful to know if the translocase domain was dispensable for localization/activity).

We are currently preparing another manuscript that documents the contribution of the various domains of FtsK to its localization profile and whether the division defect in *ftsk* knockdown cells can be suppressed by specific subdomains of FtsK. This manuscript not only will include these data, it will also include experiments that address how the site of polarized budding is defined. In the revised manuscript, we have included a description of how the domain organization of chlamydial FtsK is similar to *E. coli* FtsK (pg. 16 of revision). Chlamydial FtsK also has a similar domain organization as SpoIIIE from *B. subtilis.* The C-terminal catalytic domain of SpoIIIE is 45% identical to chlamydial FtsK. The N-terminus of SpoIIIE is predicted to encode 4 transmembrane spanning helices, like chlamydial FtsK. However, the N-terminus of SpoIIIE shares no sequence homology with the N-terminus of chlamydial FtsK. We have not included the similar domain organization of SpoIIIE and chlamydial FtsK in the revised manuscript.

(9) It seems that FtsK foci localize to a particular spot opposite from the active septum, although how this spot is specified is not clear. Is there any geometric clue for FtsK's localization like there is for Min-specified FtsZ localization?

As mentioned above, we are currently preparing another manuscript that documents our efforts to understand how the site of polarized budding is defined. This analysis is too extensive to include in this study.

(10) As mentioned in the Summary, do the authors know whether the N-terminal membrane binding part of FtsK (FtsKn) sufficient for localization/divisome assembly in Ct as it is in other species? Oullette et al. 2012 showed that FtsKn could interact with MreB in BACTH.

We are currently preparing another manuscript that documents the contribution of the various domains of FtsK to its localization profile.

(11) The previous BACTH result with MreB and FtsKn implies that this interaction is direct, yet the current data suggest that this is not the case. Can the authors comment on this? Is this due to bridging effects inherent in the BACTH system?

We have not presented any data to indicate that FtsK and MreB do not interact. We have only shown that FtsK localization is not dependent upon MreB filament formation (Fig. 3).

(12) The FtsZ-independent role of FtsK in nucleating the divisome suggests that Ct FtsK may differ from other FtsKs structurally - can this be explored, perhaps with AlphaFold 3?

As mentioned above, we have included a paragraph in the discussion of the revised manuscript (pg. 16 of the revised manuscript) that states the domain organization of chlamydial FtsK is similar to *E. coli* FtsK. This conserved domain organization is evident when we view the structures of the proteins using Alphafold.

(13) Typo on line 559: should be HeLa.

This has been corrected.